# The grid-level fixed asset model developed for China from 1951 to 2020

Danhua Xin[1,2,3], James Edward Daniell[4,5], Zhenguo Zhang[1,2,3], Friedemann Wenzel[4,5], Shaun Shuxun Wang[2,6], Xiaofei Chen[1,2,3]

[1]Department of Earth and Space Sciences, Southern University of Science and Technology, Shenzhen, 518055, China

[2]Institute of Risk Analysis, Prediction and Management (Risks-X), Academy for Advanced Interdisciplinary Studies, Southern University of Science and Technology, Shenzhen, 518055, China

[3]Key Laboratory of Earthquake Forecasting and Risk Assessment, Ministry of Emergency Management, Southern University of Science and Technology, Shenzhen 518055, China

[4]Center for Disaster Management and Risk Reduction Technology (CEDIM), Karlsruhe Institute of Technology (KIT),
Karlsruhe, 76344, Germany

[5]Geophysical Institute, Karlsruhe Institute of Technology, Karlsruhe, 76187, Germany

[6]Department of Finance, Southern University of Science and Technology, Shenzhen, 518055, China

*Correspondence to*: Zhenguo Zhang (zhangzg@sustech.edu.cn)

**Abstract.** To better aid the quick and accurate assessment of economic loss after the occurrence of future damaging earthquakes, we develop a grid-level fixed asset model for China covering the period from 1951 to 2020. The modelling process can be divided into two stages: (1) the compilation of provincial-level fixed asset data series using the perpetual inventory method (PIM) and fixed assets-related statistics; (2) the disaggregation of provincial-level fixed assets into grid-level (1 km × 1 km resolution) using different combinations of remote sensing ancillary data (i.e., nighttime light, built-up
surface area, population) for different periods, considering their temporal availability. As of 2020, the total estimated value of fixed assets in China reaches 589.31 trillion Chinese yuan (in the 2020 price level). Consistency checks have been performed by comparing our modelled fixed assets with those from other studies and data sources at different administrative levels, and good consistency has been achieved. An application to the direct economic loss estimation of the 2023 Ms6.2 Jishishan earthquake that occurred in Gansu province, China, is also presented to demonstrate the potential of the developed fixed asset
data for future damaging earthquake loss estimation in China. In the end, the limitations of the developed fixed asset model are discussed to better shed light on future improvement directions. The modelled grid-level fixed asset maps from 1951 to 2020 can be conveniently extended to more recent years as new statistics on fixed assets become available.

# 1 Introduction

As a country frequently stricken by natural hazards, China has experienced more than 355 damaging earthquakes over the past seven decades, leading to over 345,000 fatalities (Li et al., 2021) and economic losses totalling 1,437.6 billion Chinese yuan (CNY) (calculated in the price level of 2020). Meanwhile, China is also undergoing unprecedented economic, social, and urban development, with its urban population increasing from 57.7 million in 1949 to 848.4 million in 2019 (NBSC, 2020). This development process has also significantly increased the national average GDP per capita, which is around 190 times that of the early 1950s when calculated in constant prices of 2020, as shown in Figure 1. When associating socioeconomic development with natural hazards (such as earthquakes), it is evident that rapid urbanization and economic growth have significantly increased the exposure of people and fixed assets to earthquake threats (Hu et al., 2010; Yang and Kohler, 2008).

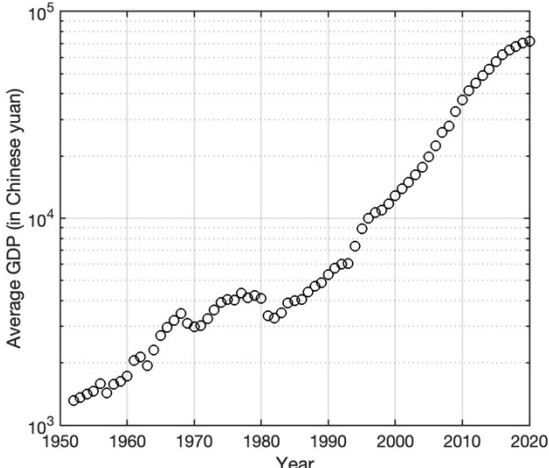

**Figure 1: The changing trend of China's national average GDP per capita from 1951 to 2020 (calculated in constant prices of 2020).**

After the occurrence of a damaging earthquake, a rapid and accurate assessment of the severity and scale of seismic fatality and economic loss is vital to assist civil protection authorities in designing post-earthquake emergency actions and allocating the search and rescue teams to the areas most needed. Specifically for seismic loss estimation, which is to translate the physical damage of buildings and structures into total monetary loss using local estimates of repair and reconstruction costs (Erdik et al., 2011), three accuracy levels (Level 1, Level 2, and Level 3) are classified in HAZUS-MH (FEMA, 2019) as differentiated by the data sources and details of exposed buildings and infrastructures integrated into the exposure model. Level 1 is a relatively rough estimation since the input data mainly include demographic data and building-related statistics extracted from the national census. Level 2 refers to a more accurate estimation, in which more detailed information on demographic data, buildings, and infrastructure information at the local level will be involved. In contrast, Level 3 corresponds to the most accurate estimation since detailed engineering inputs and specific conditions of exposed elements will be investigated in detail and employed in the estimation process. For rapid assessment of post-earthquake loss, the estimation at Level 1 is more suitable, in which the exposure models are derived mainly from demographic data, building-related statistics, and remote sensing

techniques (Erdik et al., 2010). Therefore, the fixed asset model to be developed in this paper is also based on the Level 1 data.

Besides the exposure model, empirical seismic vulnerability functions (Jaiswal and Wald, 2013) that define the seismic loss ratio as a function of macro-seismic intensity are also needed for post-earthquake rapid loss assessment. The development of empirical seismic vulnerability functions depends on damage-related information from historical earthquakes, which includes (a) macro-seismic intensity maps, (2) recorded losses of damaging earthquakes, and (c) the value of fixed assets (e.g., buildings, infrastructures, etc.) exposed to each damaging earthquake that occurred in different years. In our previous work, a composite catalog of damaging earthquakes that occurred in mainland China since 1949 (hereafter referred to as MCCDE-CAT) has been compiled (Li et al., 2021), in which the intensity maps and recorded losses have been collected for each of the damaging earthquakes. Therefore, to aid the post-earthquake rapid loss estimation work in China, in this paper, we will construct a grid-level fixed asset data series from 1951 to 2020 for China considering the availability and completeness of fixed asset-related statistics, from which the exposed asset value to damaging earthquakes in MCCDE-CAT can be extracted. Such information can be further used for the regression of empirical loss vulnerability curves following the practice in Jaiswal and Wald (2013) and Daniell (2014). The fixed asset considered in this paper includes buildings, infrastructure, and equipment, also known as the wealth capital stock (WKS). Different from the Gross Domestic Product (GDP) data, which is the standard economic indicator describing the value added through the production of goods and services in a country during a specific period, the value of the fixed asset provides the benchmark of the maximum potential direct loss of the earthquake (Wu et al., 2014), since natural hazards could cause economic losses much larger than the annual GDP (Bilham, 2010). It is noteworthy that the fixed asset value of the following year is not simply the sum of the last year's asset value and GDP.

A growing number of studies have been conducted in recent years to estimate the fixed asset value for disaster risk analysis and management at regional (Sarica et al., 2020; Wu et al., 2019), national (Kleist et al., 2006; Ma et al., 2021; Seifert et al., 2010; Thieken et al., 2006; Wu et al., 2018; Xin et al., 2021), or global scales (Daniell et al., 2011; De Bono and Chatenoux, 2015; De Bono and Mora, 2014; Dell'Acqua et al., 2013; Eberenz et al., 2020; Gamba, 2014; Gamba et al., 2012; Gunasekera et al., 2015; Jaiswal et al., 2010). However, these studies only provide the asset value for one specific year (generally the year before the publication year of these works), which cannot meet the requirement for the development of the empirical vulnerability models since fixed asset values exposed to earthquakes that occurred in different periods are needed. Unfortunately, there is no officially recorded fixed asset accumulation data in China. As an alternative, the perpetual inventory method (PIM) is considered, which was first proposed by Goldsmith (1951) and is the most frequently used method by economists to evaluate the spatial and temporal change of the macro economy of a country or region, as summarized in the review of Wu et al. (2014) for such studies conducted in China. To develop the fixed asset data series for each of the 31 provincial-level administrative units in mainland China from 1951 to 2020, the perpetual inventory method (PIM) is used in

this paper following the data compilation procedure elaborated in Zhang (2008). Notably, Hong Kong, Macao, and Taiwan are excluded from this study due to their economic and political status difference from other Chinese provinces and the lack of

necessary statistical data.

Although the temporal data series of fixed asset in China can be constructed following the PIM, if their spatial resolution is limited to the provincial level, it still cannot meet the need for accurate seismic loss estimation since spatial mismatches always exist between this level of exposure data and the extent of seismic ground shaking maps (Thieken et al., 2006). Therefore, the

provincial-level fixed asset data remains too coarse to support a reliable loss estimation for a damaging earthquake. For example, after the occurrence of the 2008 Ms 8.0 Wenchuan earthquake in Sichuan, China, most of the rescue resources (including but not limited to emergency personnel and equipment, food and medicine, tents, etc.) were sent to the less damaged city of Dujiangyan. At the same time, the Qingchuan County, one of the most severely affected areas, did not receive an appropriate rescue response. The primary reason for this problem was that the exposure data (population, buildings) used to

assess seismic loss were based on relatively rough administrative units (Xu et al., 2016). To avoid such problems and improve the spatial resolution of the exposure model in future seismic loss estimation, the provincial-level fixed asset data need to be further disaggregated into a higher resolution (e.g., 1 km × 1 km) by using appropriate ancillary information.

To perform disaggregation analyses, previous studies have employed a series of ancillary datasets derived from remotely

sensed images, such as land use and land type data (Aubrecht and León Torres, 2015; Eicher and Brewer, 2001; Elvidge et al., 2007; Hurtt et al., 2011; Liu et al., 2003), population spatial distribution datasets (Balk and Yetman, 2004; Chen et al., 2020; Freire et al., 2016; Gaughan et al., 2013; Klein Goldewijk et al., 2010; Linard et al., 2012), nighttime light data (Aubrecht and León Torres, 2016; Chen and Nordhaus, 2011; Li et al., 2020; Ma et al., 2012; Zhao et al., 2017), and road network data (Koks et al., 2019; Zhang et al., 2015; Zhu et al., 2020), to name just a few. The selection of appropriate ancillary information is

considered the most challenging part since such information should be geo-coded, readily available, and highly correlated with the exposure data to be disaggregated (Wu et al., 2018). In previous studies, socioeconomic data (e.g., GDP, fixed asset, electric power consumption, fossil fuel $CO_2$ emission, etc.) was spatially disaggregated to each pixel by assuming it is proportional to the digital number (DN) value of nighttime light images (Doll et al., 2006; Ghosh et al., 2010; Oda and Maksyutov, 2011; Zhao et al., 2011, 2012). The logic behind such practice is that a region with brighter lights at night is considered to have more

commercial and industrial activities, producing greater GDP, consuming more electricity, and emitting more $CO_2$. However, the correlation between nighttime light brightness and the amount of $CO_2$ emission was found to be exponential rather than linear by Zhao et al. (2015). Therefore, it was inferred in Zhao et al. (2017) that the correlation between the brightness of nighttime lights and the accumulation of GDP should also be exponential rather than linear. Thus, using only nighttime light data to proportionally disaggregate GDP would inevitably lead to over-distribution in suburban areas and under-distribution in

urban areas since a certain number of saturated pixels exist in nighttime light products. To solve this problem, Zhao et al. (2017) multiplied nighttime light images with LandScan population data to produce the lit-population (hereafter referred to as "lit-pop") images. They used the lit-pop value as the weight indicator to disaggregate China's administrative-level GDP datasets. On the one hand, this is because the correlation between the DN value of nighttime light and population is also exponential. On the other hand, integrating population data into the disaggregation process can help overcome the saturation problem of nighttime light data since the range of DN values is limited to 0 - 63. For suburban areas where the DN values are relatively small, the population density also increases slowly, and saturation is not a problem; however, with progressively higher DN values, the increase in population density will also grow rapidly and finally lead to a considerable population density in urban core areas with a DN value of 63. The rapidly increased asset value in such areas can thus be better represented by lit-pop than by nighttime DN value or population alone (Zhao et al., 2017).

As emphasized by Zhao et al. (2017), the lit-pop indicator they produced has no measurement unit. It represents neither people count nor nighttime light brightness in real life, but rather the economically weighted population. Compared with using nighttime light or population data alone, the use of lit-pop as the economic indicator can better reflect the spatial heterogeneity of the economy. This is because when two regions have the same population but different DN values of nighttime light, the region with a higher DN value has a larger lit-pop and consequently has a larger distributed GDP than the one with dimmer nighttime light. Based on the lit-pop approach in Zhao et al. (2017), Eberenz et al. (2020) generated a globally consistent gird-level asset exposure dataset for 224 countries, in which the unwanted artifacts (including blooming, saturation, and lack of detail) are mitigated by using a combination of nightlight and population data. The GDP comparison for 14 countries by Eberenz et al. (2020) also showed that the disaggregation effect using nighttime lights or population data alone is not as good as using their combination. Inspired by the work of Zhao et al. (2017) and Eberenz et al. (2020), in this study, the provincial-level fixed asset data will be further disaggregated into grid level based on the combined use of nighttime light, population, and other available supplementary data (e.g., built-up surface area data), to generate the final grid-level fixed asset data series for 31 provinces in mainland China during 1951-2020.

The main structure of the following sections in this paper will be organized as follows. Section 2 will detail the data and methods used to compile the provincial-level fixed assets and explain how to disaggregate them into the grid level. In Section 3, the modelled fixed assets at the provincial level during 1951-2020 will be presented, and the grid-level fixed asset map for 2020 will also be demonstrated. Furthermore, the temporal change and spatial distribution characteristics of grid-level fixed assets in China's three largest urban agglomerations will be demonstrated and compared. Section 4 will examine the consistency between disaggregation indexes used in different periods. The consistency of our modelled fixed asset data with those developed by other studies will also be evaluated at different administrative levels. Limitations of the current study will

be discussed as well to better outline future improvement directions.

## 2 Data and Methods

The data and methods will be introduced separately for the two parts of contents involved in this paper: (1) the compilation of provincial-level fixed asset data for 31 provinces in China from 1951 to 2020, and (2) the disaggregation of the provincial-level fixed asset data into 1 km × 1 km grids using different weighting indicators for different periods. The flow chart to be followed in the modelling process is shown in Figure 2. The datasets to be used are summarized in Table 1. A detailed introduction of the data inputs and the modelling steps will be given in the following sections.

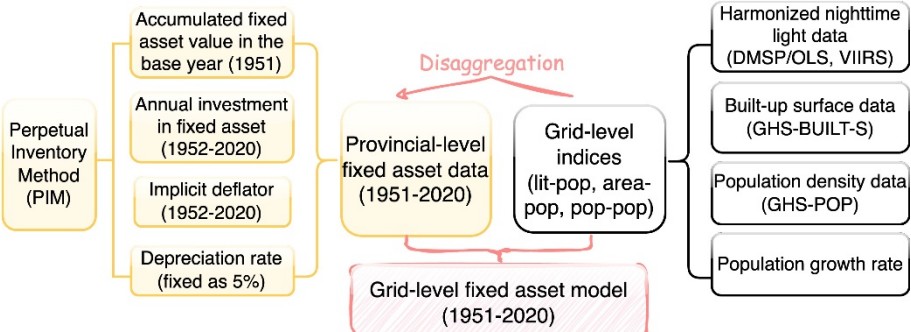

**Figure 2: The flow chart followed to develop the fixed asset model in this paper. Boxes are marked by different colors to differentiate the two main parts in the workflow. The boxes marked in yellow refer to the steps required to develop the provincial-level fixed asset data, and the boxes marked in white are components used to construction the disaggregation indices (see Section 2 for more details). The indices "lit-pop, area-pop, pop-pop" refer to the disaggregation index generated using nighttime light and population, build-up area and population, and population data alone, respectively. The abbreviation "DMSP/OLS" refers to nighttime light observations acquired by the US Air Force Defense Meteorological Satellite Program (DMSP) Operational Linescan System (OLS), "VIIRS" refers to nighttime light observations from the Visible Infrared Imaging Radiometer Suite instrument, "GHS-BUILT-S" refers to the built-up surface data provided by the Global Human Settlement Layer project, and "GHS-POP" refers to the population density data provided by the same project.**

**Table 1: A summary of datasets to be used in this paper.**

| Usage | Data type | Year range | Spatial resolution | Description |
|---|---|---|---|---|
| For provincial-level fixed asset modelling in China | Accumulated fixed asset value in the base year | 1951 | Provincial-level | Section 2.1.1 |
| | Annual investment in fixed asset | 1952-2020 | | Section 2.1.2 |
| | Implicit deflator | 1952-2020 | | Section 2.1.3 |
| | Depreciation rate (fixed as 5%) | 1951-2020 | | Section 2.1.4 |
| For grid-level fixed asset modelling in China | Harmonized nighttime light data from DMSP/OLS and VIIRS | 1992-2020 | 1km×1km | Section 2.2.1 |
| | Built-up surface data | 1975-2020 (in 5-year | | Section |

| | | intervals) | | 2.2.2 |
|---|---|---|---|---|
| | Population density data | 1975-2020 (in 5-year intervals) | | Section 2.2.2 |
| | Population growth rate data | 1951-2020 | Provincial-level | Section 2.2.3 |

## 2.1 Construction of the provincial-level fixed asset data by using the perpetual inventory method (PIM)

To construct China's provincial-level fixed asset data during 1951-2020 using the PIM, four types of information need to be determined, namely (1) the value of the accumulated fixed asset in the base year, (2) the annual fixed asset investment in each province, (3) the implicit deflator of fixed asset, and (4) the depreciation rate or service life of the fixed asset. Assuming the efficiency of the fixed asset follows a geometric diminishing model (Wu et al., 2014; Zhang, 2008), for each province, the accumulated fixed asset value at year $t$ (namely $K_t$) is defined as follows:

$$K_t = K_{t-1}(1 - \delta_t) + I_t, \ t \in [1952 \ 2020] \tag{1}$$

Where $\delta_t$ is the depreciation rate of fixed asset, and $I_t$ is the total investment in fixed asset (TIFA) at year $t$.

### 2.1.1 Determination of the accumulated fixed asset value of the base year 1951

Since the foundation of the People's Republic of China in 1949, the currency was then uniformly switched to the Chinese yuan (CNY) and 1949 could be set as the base year. However, due to the lack of large-scale surveys or census on TIFA in 1949 and 1950 for many of the 31 provinces, 1951 is selected as the base year in this paper. As adopted by previous studies, one method to estimate the accumulated fixed asset value in the base year is by referring to the capital-output ratio method (e.g., Zhang, 1991; Chow, 1993; Perkins, 1988; He et al., 2003), in which the value of the accumulated fixed asset is set to be around three times of the GDP in the base year. Another way to roughly approximate the accumulated fixed asset value is by dividing the fixed capital formation (FCF) of the base year by the sum of the long-run growth rate of investment (e.g., the constant-price FCF) and the depreciation rate (Hall and Jones, 1999; Wu et al., 2014; Zhang, 2008). In this paper, following the practice in several previous studies (e.g., Zhang, 1991; Chow, 1993; He et al., 2003), each province's accumulated fixed asset value in 1951 is determined by multiplying their TIFA in 1951 by 50 times. The estimated overall value of the accumulated fixed asset of the base year in China is around 94.9 billion Chinese yuan (in the price level of 1951), which corresponds to 2343.5 billion Chinese yuan when adjusted to the 2020 price level. The determination process of the initially accumulated fixed asset value for the base year has inevitable uncertainty. Fortunately, previous studies (Shan, 2008; Wu et al., 2014; Zhang et al., 2004) have demonstrated that the effect of the initially determined fixed asset value of the base year on the asset estimation for the following years will decline given sufficiently long time series. For example, the sensitivity test performed by Wu et al. (2014) indicated that a doubling of the initial asset value in 1978 only resulted in less than a 0.6% change in the stock estimation in

2012.

### 2.1.2 Collection of annual investment data in fixed asset

To get a complete data series of the annual total investment in fixed asset (TIFA) for each province during 1951-2020 (namely $I_t$ in Eq. (1)), we first refer to the book *China Compendium of Statistics 1949-2008* compiled by Department of Comprehensive Statistics of National Bureau of Statistics (DCSNBS, 2009), in which the annual investment data in fixed asset at both national level and provincial level were given in the price level of each year up to 2008. The TIFA data in 1951 for Hainan and Tibet are not available in the reference mentioned above. Instead, we assign it to be 0.01 and 0.0002 billion Chinese yuan (in 1951

price level), around 50% of the TIFA in 1952 for Hainan and Tibet, respectively. To complement this data series for years after 2008, we further refer to the use of the TIFA data records in the yearly statistical books of China from 2009 to 2020.

### 2.1.3 Compilation of the implicit deflator of the fixed asset

     To calibrate the deflation of TIFA with time, we convert the TIFA values given in the price level of each year to the constant price of the reference year by using the "price index of fixed asset", which is also called the implicit deflator of the fixed asset.

Theoretically, the calculation of the implicit deflator should be based on the weighted average of the price indexes for each of the three components of fixed asset investment (namely investment on construction and installation, purchases of equipment and instruments, and others), with their weight determined by the asset percentage of each component for each province during 1951-2020. However, due to the lack of related statistics, we refer to using the provincial-level investment data to derive the price index of each province. According to Wu et al. (2014) and Zhang (2008), the formula to derive the implicit deflator

(namely $Ide_t$) can be expressed as follows:

$$Ide_t = \frac{FCF_t}{FCF_{t-1} \times FCF\_index_t}, \ t \in [1952 \ 2004] \tag{2}$$

The detailed derivation process of this formula was given in Zhang (2008). Ideally, $FCF_t$, $FCF_{t-1}$ should be the fixed capital formation (FCF) in year $t$ and $t-1$, respectively. However, at the provincial level, the FCF data before 1978 are not publicly accessible. Therefore, we use TIFA to replace FCF when calculating $Ide_t$. On the one hand, TIFA is more often used and

investigated in China; on the other hand, FCF and TIFA have similar dynamic changing trends (Qin et al., 2006). $FCF\_index_t$ refers to the gross fixed capital formation growth rate in year $t$ calculated in constant price (previous year = 1). The $FCF\_index_t$ data for the years 1952-2004 can be found in the book *Data of Gross Domestic Product of China 1952-2004* compiled by the Department of National Accounts of National Bureau of Statistics of China (DNANBSC, 2007). For years after 2004, the implicit deflator ($Ide_t$) can be replaced by the price index of the fixed asset comprehensively compiled from

the book *China Compendium of Statistics 1949-2008* (DCSNBS, 2009) for years 2005-2008, from the official website of the National Bureau of Statistics (https://data.stats.gov.cn) for years 2009-2019, and from Tables 5-7 of *China Statistics Yearbook*

*2021* for the year 2020. Notably, in some provinces the $FCF\_index_t$ data are incomplete. In this case, $FCF\_index_t$ data from neighboring provinces are used to compensate for the missing information. For example, the FCF growth rate data of Chongqing for the years 1952-1997 are taken from the data of Sichuan province since Chongqing belonged to Sichuan before 1997 and was set as a municipality directly under the reign of the central government of China afterward. The missing growth rates of FCF during 1952-1977 of Guangdong province are taken from the data of Guangxi province since they are geographically close. For the same reason, the missing data of Tibet from 1952 to 1992 are supplemented by the average of the FCF growth rate data of Qinghai and Xinjiang provinces. Since Hainan was not a province until 1997, the $FCF\_index_t$ data during 1952-1996 of Hainan are taken from its neighboring Guangdong province.

### 2.1.4 Determination of the depreciation rate of the fixed asset

The consideration of depreciation of the fixed asset is necessary when estimating the asset value of accumulated capital stock in previous years, which will diminish over time. In earlier studies, the depreciation rate was usually set as a fixed value within the range of 5%~10% (e.g., Perkins, 1988; Hall and Jones, 1999; Wang and Yao, 1999). In Zhang (2008), the depreciation rate of fixed asset was determined by considering the service life ($T$) of different fixed asset types (including but not limited to construction and installation, equipment and instrument) and their residual value ($d_T$) when the capital goods are retired. The calculation formula of the depreciation rate ($\kappa$) is defined as follows:

$$d_T = (1 - \kappa)^T \tag{3}$$

In Zhang (2008), the service life ($T$) of construction and installation, equipment and instruments, and other types of fixed asset in China was set as 45 years, 20 years, and 25 years, respectively. Their residual value ($d_T$) was uniformly set as 4%. The depreciation rates of these three fixed asset types were calculated to be 6.9%, 14.9%, and 12.1%, respectively. Ideally, the relative weights of each of the three fixed asset types should also be considered to determine a comprehensive depreciation rate of the fixed asset. However, due to a lack of such data at the provincial level, the weight at the national level was used in Zhang (2008), which is 63% for completion of construction and formation, 29% for purchase of equipment and instruments, and 8% for other investments. Finally, under the assumption of geometric diminishing of the relative efficiency, the comprehensive deprecation rate of the fixed asset was determined to be 9.6% for the whole nation. Following the method in Zhang (2008), Wu et al. (2014) calculated the depreciation rate range for each of the 31 provinces in mainland China based on newly released composition data of TIFA for each province and by setting the residual value of the fixed asset to be 3%~5% of their original value. Their provincial-level depreciation rate of the fixed asset is within the range of 7.95%~10.05%. The comparison analysis of Li (2011) indicates that the change of 1% in depreciation rate will lead to a 10% change in accumulated fixed asset 25 years later. Li (2011) also suggested that the depreciation rate should be within the range of 5%~10%. In this paper, since the development of provincial-level fixed asset data is to be used for rapid emergency response after the occurrence

of damaging earthquakes in China, the replacement values of different types of fixed asset in earthquake-affected areas are generally higher than their residual values even they have lasted for a much longer time than their service lives; therefore, a conservative depreciation rate of 5% is chosen for all provinces to get the final accumulated fixed asset data series from 1951

to 2020. Currently, all the data inputs required for estimating accumulated fixed asset values are ready. The provincial-level fixed asset data from 1951 to 2020 can be constructed following Eq. (1). This dataset will be demonstrated and evaluated in the Results and Discussion section. Next, we will introduce the disaggregation method used in this paper to distribute the provincial-level fixed assets into the grid level.

## 2.2 Disaggregation of the provincial-level fixed asset into 1km × 1km grids

Given the exponential relation between population/nighttime light and socioeconomic data (as explained in detail in the Introduction section), to disaggregate the provincial-level fixed assets into 1 km × 1 km grids, nighttime light (available since 1992) and population density data (available since 1975) will be combined to generate the lit-pop index. For years before 1990, the lack of nighttime light data will be compensated by built-up surface area data (available since 1975) to create the area-pop index. For years before 1975, the spatial distribution of population density data will be derived from the population density

map in 1975 and the annual growth rate data for each province dating back to 1951. Then, the population density data alone will be used to create the pop-pop index during 1951-1970. More information on these datasets and the creation process of the disaggregation indexes will be introduced in the following sections.

### 2.2.1 The nighttime light data

The nighttime light data from 1992 to 2020 with a spatial resolution of 30 arc-seconds  (around 1000m at the Equator) and DN

values ranging from 0 to 63 are compiled by Li et al. (2020). In Li et al. (2020), an integrated and consistent nighttime light dataset at the global scale was compiled by harmonizing the intercalibrated nighttime light observations acquired by the US Air Force Defense Meteorological Satellite Program (DMSP) Operational Linescan System (OLS) (hereafter referred to as the DMSP/OLS data) during 1992-2013 and the simulated DMSP/OLS-like nighttime light observations from the Visible Infrared Imaging Radiometer Suite instrument ( hereafter referred as the VIIRS data) during 2014-2020. The original DMSP/OLS data

were recorded by six different satellites during 1992-2013 with a spatial resolution of 30 arc-seconds and a near-global coverage of 180°W to 180°E in longitude and 65°S to 75°N in latitude (Zhao et al., 2019). Inconsistency exists between these original DMSP/OLS data due to the lack of onboard calibration, satellite shift, varied atmospheric conditions, sensor degradation, etc. Therefore, a stepwise calibration approach was performed by Li et al. (2020) before harmonizing the DMSP/OLS data with VIIRS data. Unlike the annual DMSP/OLS data, the VIIRS data have been available since 2013. They

are recorded monthly with an improved radiometric resolution and a spatial resolution of 15 arc-seconds across the latitudinal zone of 65°S-75°N (Miller et al., 2012). In the monthly recorded VIIRS data, errors due to bio-geophysical processes (e.g.,

seasonal dynamics of vegetation and snow) were corrected, and observations affected by stray light were excluded. These monthly records were further pre-processed and combined into annual time series data using the weighting average approach and finally converted into DMSP/OLS-like nighttime light observations using a sigmoid function initially proposed by Zhao et al. (2020) in Southeast Asia. The DMSP/OLS-like nighttime light converted from the original VIIRS data has been available and updated since 2014 by Li et al. (2020).

### 2.2.2 The population density and built-up surface data

The population datasets used in this paper are provided by the Global Human Settlement Layer (GHSL) project of the Joint Research Centre, European Commission (Freire et al., 2016; Schiavina et al., 2022b), which have been available in 5-year intervals since 1975 (hereafter referred to as GHS-POP). The number of people per grid (with resolutions ranging from 2m to 1km) is given in each GHS-POP raster file, which was disaggregated from the raw global census data harmonized for the Gridded Population of the World (GPW) by CIESIN (Freire et al., 2015) and the proxy used in this disaggregation process was the built-up density mapped in the GHSL global layers per corresponding epoch (Maffenini et al., 2023). Compared with its previous version, major improvements of the datasets are the following: use of built-up volume maps (abbreviated as GHS-BUILT-V R2022A); use of more recent and detailed population estimates derived from GPWv4.11 integrating both *UN World Population Prospects 2022* country population data (Gaigbe-Togbe et al., 2022) and *World Urbanisation Prospects 2018* data (UNDESA, 2018) on cities; revision of GPWv4.11 population growth rates by convergence to upper administrative level growth rates; systematic improvement of census coastlines; systematic revision of census units declared as unpopulated; integration of non-residential built-up volume information (abbreviated as GHS-BUILT-V_NRES R2023A); spatial resolution of 100m Mollweide and 3 arcseconds in WGS84 projection system; projections to 2030.

The built-up surface data used to allocate the GHSL population information are also provided in 5-year intervals between 1975 and 2030 (hereafter referred to as GHS-BUILT-S). They are generated by spatial-temporal interpolation of five observed collections of multiple-sensor, multiple-platform satellite imageries, namely the Landsat (MSS, TM, ETM sensor) supporting 1975, 1990, 2000, and 2014 epochs, and the Sentinel-2 (S2) composite (GHS-composite-S2 R2020A) supporting the 2018 epoch (Pesaresi and Politis, 2022; Schiavina et al., 2022a). In addition, the research findings of the world settlement footprint suite launched by the German Aerospace Centre (DLR) in collaboration with the European Space Agency (ESA) and the Google Earth Engine team (Marconcini et al., 2021) are also integrated into the development process of the GHS-BUIT-S dataset.

### 2.2.3 The lit-pop, area-pop, and pop-pop index series

Following the practice in Eberenz et al. (2020), the lit-pop index is created from the combination of nighttime light and

population data, with its definition given in the following Eq. (4):

$$Lit^n Pop^m_{grid} = (NL_{grid} + \delta)^n \cdot Pop^m_{grid} \tag{4}$$

In each grid, the value of the disaggregation index ($Lit^n Pop^m_{grid}$) is the product between the DN value of the nighttime light

image ($NL_{grid}$) ranging from 0 to 63 and the number of population ($Pop_{grid}$). When $Pop_{grid} > 0$, the value of $\delta$ is set as 1 to ensure that the lit-pop value of non-illuminated but populated grids will not get zero (Eberenz et al., 2020). In cases when $Pop_{grid} = 0$, $\delta$ is set as 0, and nighttime light data alone are used to represent the fixed asset share. To evaluate the performance of this disaggregation methodology, Eberenz et al. (2020) conducted the performance evaluation tests by applying 10 different combinations of $m$ and $n$. Their tests showed that the disaggregation performance would be the best when $m$ and

$n$ were set as 1. Therefore, in this paper the values of $m$ and $n$ in Eq. (4) are also set as 1.

The nighttime light data are only available from 1992. Assuming the nighttime light will not change too much between 1991 and 1992, while for years before 1991, new ancillary information needs to be employed to create the quasi lit-pop index. The built-up surface data developed by the GHSL project of the Joint Research Centre, European Commission (hereafter referred

to as the GHS-BUILT-S data) are chosen for this purpose. The GHS-BUILT-S data are combined with the GHS-POP data to generate the area-pop index, which is defined in the following Eq. (5):

$$Area^n Pop^m_{grid} = (Area_{grid} + \delta)^n \cdot Pop^m_{grid} \tag{5}$$

Where $Area_{grid}$ represents the built-up area in each grid, and the definitions of $Pop_{grid}$, $\delta$, $m$, and $n$ are same as those in Eq. (4).

Unfortunately, the GHS-BUILT-S and GHS-POP data are available only after 1975 in 5-year intervals. It is further assumed that the built-up surface area in China remained unchanged from 1971 to 1975 since economic activities almost ceased during this period due to the Cultural Revolution. For years before 1971, the GHS-POP data in 1975 and the provincial-level population growth rates compiled from statistical yearbooks are used to derive the grid-level population dataset from 1951 to

335 1970. Then, the derived grid-level population data alone are used to generate the pop-pop index, with its definition being given in the following Eq. (6):

$$Pop^n Pop^m_{grid} = (Pop_{grid} + \delta)^n \cdot Pop^m_{grid} \tag{6}$$

Where the definitions of $Pop_{grid}$, $\delta$, $m$, and $n$ are the same as those in Eq. (4).

It is worth noting that since the grid-level population density data provided by GHSL from 1975 are also given in 5-year intervals, the population density maps for the intervening years are derived from the compiled growth rate data and the

reference year population density map. For example, the 1 km × 1 km population maps for 1971-1974 are derived from the GHSL-issued population density map in 1975 and the compiled provincial-level population growth rate data for 1971-1974.

## 3 Results

In the Data and Methods section, the data inputs and the methods used to construct the provincial-level fixed asset data during 1951-2020 and the disaggregation process of the provincial-level fixed asset data into grid level have been described in detail using different ancillary information with varying temporal availability. To summarize, the nighttime light data and GHS-POP data are used to generate the lit-pop disaggregation indexes from 1991 to 2020, the GHS-BUILT-S data and GHS-POP data are used to construct area-pop disaggregation indexes from 1971 to 1990, and the population density data are used to derive the pop-pop disaggregation indexes from 1951 to 1970. In this section, we will first demonstrate the modelled fixed asset data for 31 provinces from 1951 to 2020. Then, the spatial-temporal characteristics of the grid-level fixed asset model in 2020 and for China's three largest urban agglomerations will be demonstrated and analysed.

### 3.1 Modelled provincial-level fixed assets from 1951 to 2020

Based on the estimation of the accumulated value of fixed asset in the base year of 1951, the compilation of the annual total investment in fixed asset (TIFA) data, the depreciation rate, and the derivation of implicit deflator data series in section 2.1, the accumulated fixed asset model can be obtained for all 31 provincial-level administrative units in mainland China from 1951 to 2020 (Figure 3). By 2020, the total estimated value of fixed asset in China reaches 589.31 trillion Chinese yuan (in the 2020 price level). Shandong and Jiangsu have the highest accumulation of fixed assets, amounting to 50.12 and 49.36 trillion Chinese yuan, respectively. Tibet and Ningxia have the lowest accumulated fixed assets, with 1.51 and 3.01 trillion Chinese yuan, respectively. In comparison, the provincial GDP ranking in 2020 issued by the Chinese government shows that Guangdong and Jiangsu have the highest GDP of 11.12 and 10.28 trillion Chinese yuan, respectively. The GDPs of Tibet and Qinghai are the lowest, at 0.19 and 0.30 trillion Chinese yuan, respectively. When further calculating the ratios between the accumulated fixed assets and GDP, as illustrated in Figure 4, their ratios differ among provinces and change across temporal periods. Therefore, it may introduce significant uncertainties in seismic loss estimation if the accumulated fixed assets are derived by multiplying the GDP by a stationary exposure correction factor, as done in some previous studies (Chen et al., 1997; Jaiswal and Wald, 2013; Sarica and Pan, 2022; Wang et al., 2009), although this method is quite convenient.

It is also noteworthy that in Figure 4, there are two abnormally high fixed asset/GDP ratios in the 1960s, which are 26 for Anhui province in 1962 and 120 for Ningxia province in 1963. For Anhui province, this is related to its exceptionally high

fixed assets in 1962, as indicated in Figure 3. For Ningxia province, this is related to its abnormally low GDP in 1963, which

is only 0.01 billion Chinese yuan (in the price level of 1963) and around 1/40 of its neighboring years, as recorded in Table

31-4 of DCSNBS (2009). For some provinces (Tibet, Guizhou, etc.), as shown in Figure 4, fixed asset/GDP ratios are lower

than one before the 1980s. This can be explained by the rough estimation made in the determination process of the initially

accumulated fixed assets as well as the lack of an official and standard method in the compilation of economic indicators in

the early periods after 1949, which also leads to the irregularly intertwined asset-changing trends modelled for different

provinces in Figure 3.

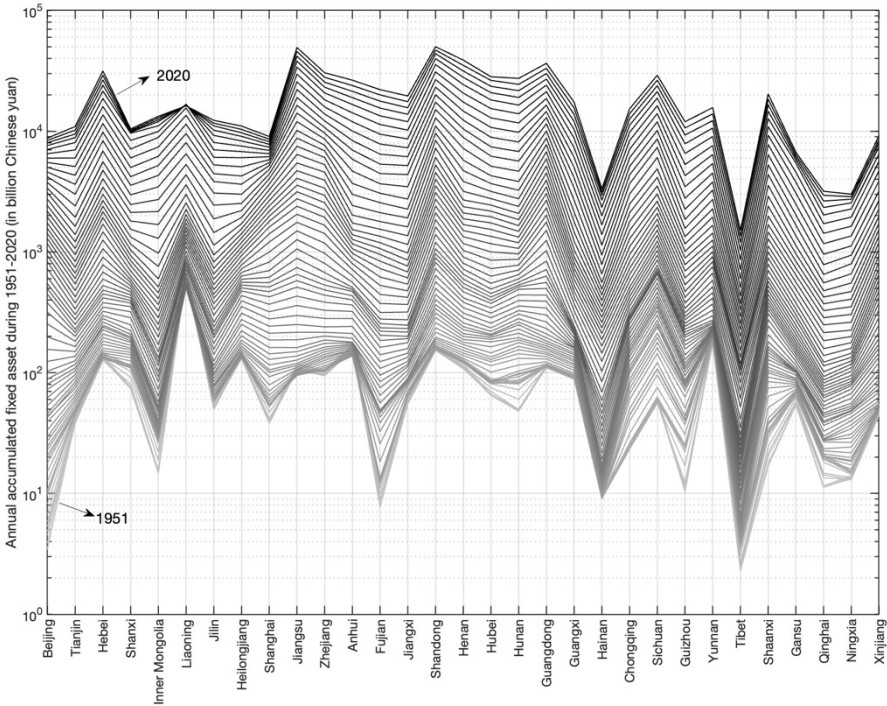

**Figure 3:The accumulated fixed asset data modelled for 31 provincial-level administrative units in mainland China during 1951-**

**2020. It is worth noting that the asset value is calculated in the constant price level of 2020. The bottom line marks the accumulated**

**fixed asset in year 1951 and the top line marks the accumulated fixed asset in year 2020.**

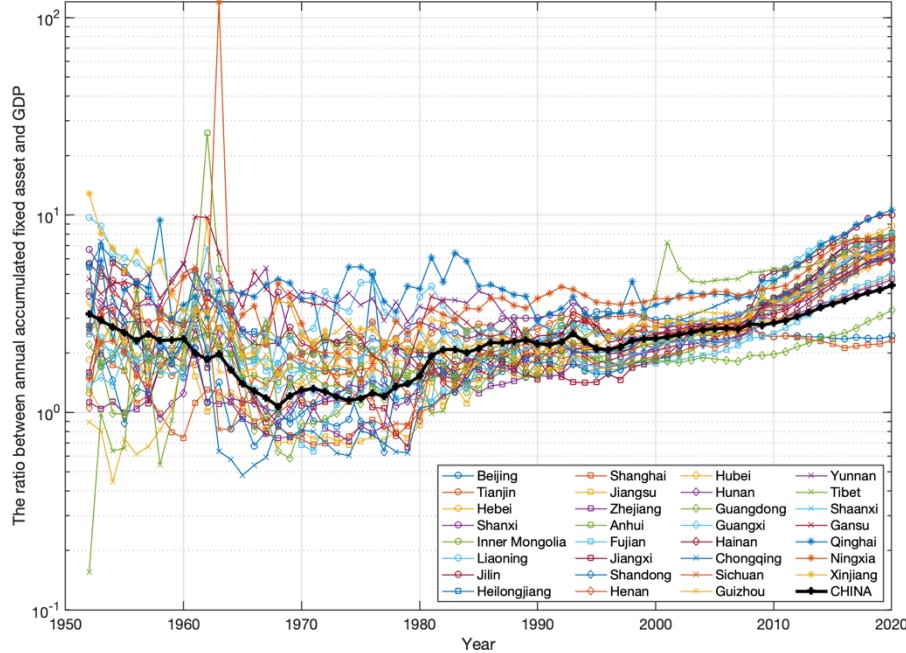

**Figure 4:The ratio between accumulated fixed assets and GDP for each province and China as a whole from 1952 to 2020.**

By using different ancillary datasets to generate the disaggregation indexes, the provincial-level fixed asset data shown in Figure 3 can be further downscaled into 1 km × 1 km asset maps. The spatial distribution map of the grid-level fixed assets in 2020 is shown in Figure 5. The locations of the capital cities of China's 31 provincial-level administrative units considered in this paper are also shown. In Figure 5, it is not surprising to observe that all 31 capital cities are in clusters of highly

accumulated fixed assets, which indicates their attractiveness to personnel and capital from their neighboring regions. As divided by the "Hu Huanyong" line (Hu, 1935), China is divided into East China and West China according to their differences in population, geography, social development, and ecological environment. As expected, the fixed assets are highly agglomerated in East China, accounting for 86% of the total asset value, which further indicates significant disparity and spatial heterogeneity in economic development within China. Compared with exposure models given by administrative units, the

grid-level fixed asset model can better help improve the accuracy of seismic loss assessment when further combined with hazard maps at varying resolutions, thus better serving the allocation needs of emergency response and risk mitigation resources.

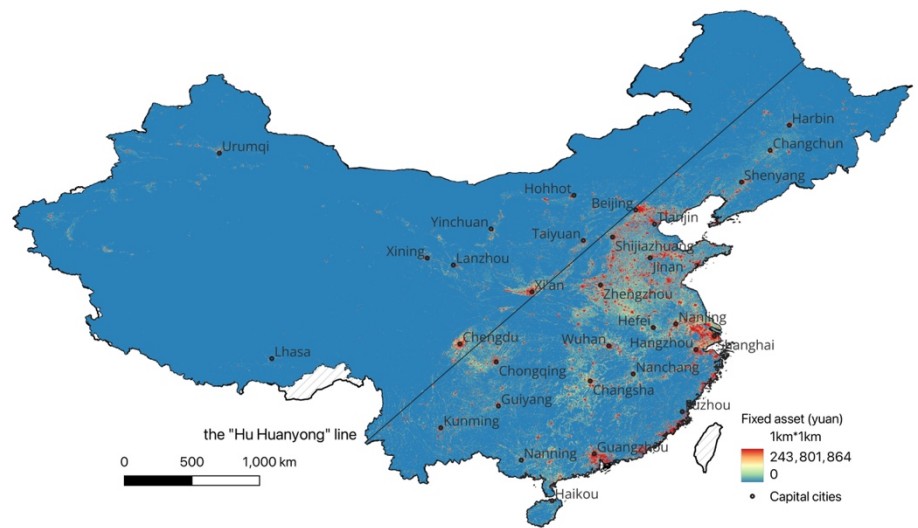

**Figure 5: The spatial distribution of our modelled grid-level fixed asset map for mainland China in 2020. The unit of the asset value is the Chinese yuan. The locations of the capital cities of 31 provincial-level administrative units are also shown. The "Hu Huanyong" line divides China into East China and West China.**

## 3.2 Spatial-temporal characteristics of fixed assets in China's three largest urban agglomerations

As shown in Figure 5, most fixed assets are clustered in East China, especially within the three largest national urban agglomerations of China (with their locations outlined in Figure 6), namely the Beijing-Tianjin-Hebei Urban Agglomeration (BTH-UA), the Yangtze River Delta Urban Agglomeration (YRD-UA), and the Pearl River Delta Urban Agglomeration (PRD-UA). The BTH-UA is composed of 13 cities, including Beijing, Tianjin, and 11 cities in Hebei province (Baoding, Cangzhou, Chengde, Handan, Hengshui, Langfang, Qinhuangdao, Shijiazhuang, Tangshan, Xingtai, and Zhangjiakou). The YRD-UA is composed of 27 cities, including Shanghai, 8 cities in Anhui province (Anqing, Chizhou, Chuzhou, Hefei, Ma'anshan, Tongling, Wuhu, and Xuancheng), 9 cities in Jiangsu province (Changzhou, Nanjing, Nantong, Suzhou, Taizhou, Wuxi, Yancheng, Yangzhou, and Zhenjiang), and 9 cities in Zhejiang province (Hangzhou, Huzhou, Jiaxing, Jinhua, Ningbo, Shaoxing, Taizhou, Wenzhou, and Zhoushan). In contrast, the PRD-UA comprises only 9 cities in Guangdong province, including Guangzhou, Shenzhen, Dongguan, Foshan, Huizhou, Jiangmen, Zhaoqing, Zhongshan, and Zhuhai. In terms of land area, the BTH-UA, the YRD-UA, and the PRD-UA are 218.0, 211.7, and 42.2 thousand square kilometers, accounting for 2.27%, 2.21%, and 0.44% of the total land area of China, respectively.

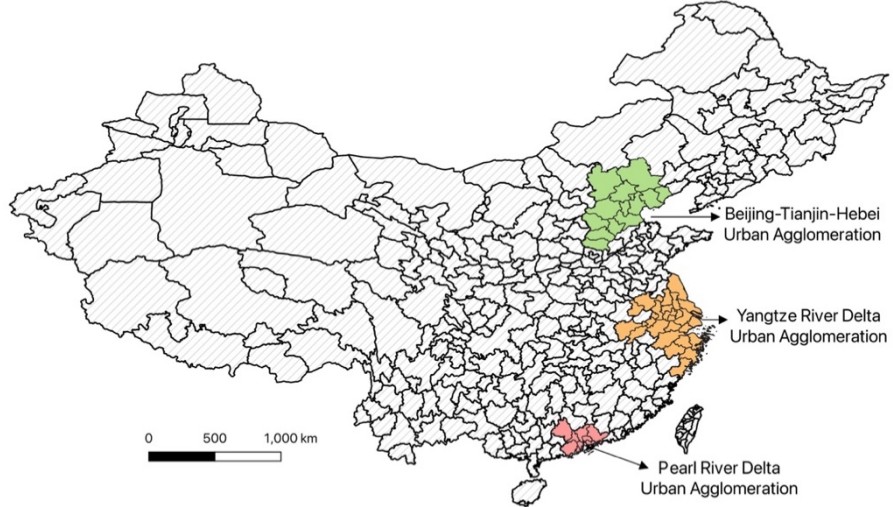

**Figure 6: The spatial locations of China's three largest urban agglomerations.**

As summarized in Table 2, the accumulated fixed assets in each agglomeration have increased over the years. Still, their changing trends of fixed asset share relative to the whole country are quite different. The fixed asset share of the BTH-UA has remained almost unchanged over the past seven decades, ranging from 9.05% in 1951 to 8.7% in 2020. Meanwhile, the fixed asset share of the YRD-UA has increased from 7.64% in 1951 to 15.33% in 2020. The increase in the fixed asset share of the PRD-UA is the largest, rising from 0.98% in 1951 to 4.01% in 2020, reflecting this region's strong economic vitality.

**Table 2: The fixed assets in China's three largest urban agglomerations. BTH-UA, YRD-UA, and PRD-UA are the abbreviations of the Beijing-Tianjin-Hebei Urban Agglomeration, the Yangtze River Delta Urban Agglomeration, and the Pearl River Delta Urban Agglomeration, respectively. Note that the fixed asset is calculated in the price level of each corresponding year.**

| Year | Fixed assets (in billion Chinese yuan) | | | | Fixed asset ratio relative to the whole country | | |
|------|--------|--------|--------|-----------------|--------|--------|--------|
| | BTH-UA | YRD-UA | PRD-UA | the whole China | BTH-UA | YRD-UA | PRD-UA |
| 2020 | 51273.59 | 90348.19 | 23659.06 | 589314.93 | 8.70% | 15.33% | 4.01% |
| 2010 | 13099.28 | 24146.34 | 6432.49 | 132783.60 | 9.87% | 18.18% | 4.84% |
| 2000 | 2507.67 | 4277.89 | 1421.02 | 21793.97 | 11.51% | 19.63% | 6.52% |
| 1990 | 327.31 | 497.83 | 103.42 | 3592.58 | 9.11% | 13.86% | 2.88% |
| 1980 | 76.00 | 85.55 | 13.94 | 772.63 | 9.84% | 11.07% | 1.80% |
| 1970 | 27.52 | 26.47 | 3.27 | 309.69 | 8.89% | 8.55% | 1.06% |
| 1960 | 34.44 | 39.64 | 5.56 | 345.83 | 9.96% | 11.46% | 1.61% |
| 1951 | 8.55 | 7.22 | 0.93 | 94.42 | 9.05% | 7.64% | 0.98% |

To better visualize the spatial changes in fixed assets, the grid-level fixed asset maps for the years 1951, 1960, 1970, 1980, 1990, 2000, 2010, and 2020 are shown in Figure 7, Figure 8, and Figure 9 for BTH-UA, YRD-UA, and PRD-UA, respectively. It is noteworthy that the fixed assets shown in Figure 7 - Figure 9 have been adjusted to the 2020 constant price level using the implicit deflator series at the national level compiled in Section 2.1.3, thus avoiding the effect of price changes on the evolution of the spatial distribution characteristics of fixed assets. To better reveal the increase of fixed assets in space over time, the

legends within each panel of Figure 7 - Figure 9 are the same, as separately determined by the value ranges of accumulated fixed assets in 1980 for these three agglomerations. For better visualization effect, the upper and lower thresholds in each legend refer to the 98%-quantile and 2%-quantile of the asset values in corresponding map.

The spatial distribution characteristics of fixed assets over the past seven decades can be divided into two periods: before the 1980s and after the 1980s. Before the 1980s, the fixed assets were mainly clustered in the big cities of each agglomeration, namely Beijing and Tianjin in the BTH-UA (Figure 7), Shanghai in the YRD-UA (Figure 8), and Guangzhou in the PRD-UA (Figure 9). The increase in clustered fixed assets before the 1980s can also be observed in other cities, but it is sparse in space and slow in speed. In contrast, fixed assets experienced a rapid and extensive increase after the 1980s, closely related to the national policy "Reform and Opening-up" issued in 1978, after which the focus of the Communist Party and the state shifted to economic development. When calculated in the 2020 constant price level, the values of accumulated fixed assets in 2020 are 76, 119, and 192 times the 1980 fixed asset values in BTH-UA, YRD-UA, and PRD-UA, respectively. Compared with the situation in 1951, the values of fixed assets in 2020 are 243, 506, and 1035 times the 1951 fixed asset values in BTH-UA, YRD-UA, and PRD-UA, respectively. This not only indicates the overall rapid accumulation speed of fixed assets in these three agglomerations after the 1980s, but also reflects the even faster growth rate in the PRB-UA compared to the BTH-UA and the YRD-UA. This comparison further reveals the extraordinarily high economic dynamism in the PRD-UA.

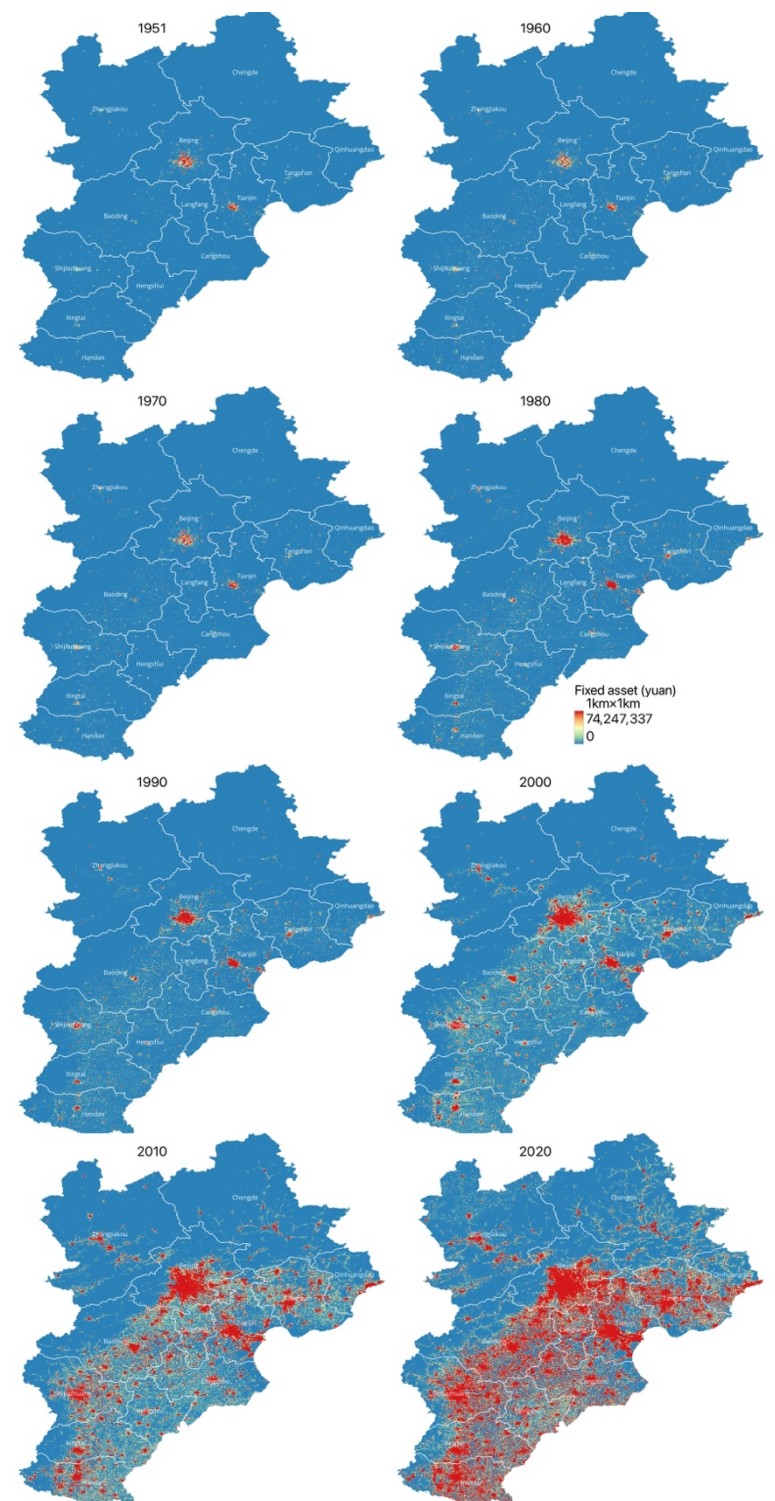

**Figure 7: The spatial distribution maps of grid-level fixed assets (adjusted to 2020 constant price level) modelled for the Beijing-Tianjin-Hebei Urban Agglomeration (BTH-UA) in 1951, 1960, 1970, 1980, 1990, 2000, 2010, and 2020. The legend is generated based on the value range of fixed assets in 1980 and is uniformly applied to the asset maps of other years for better visualization effect.**

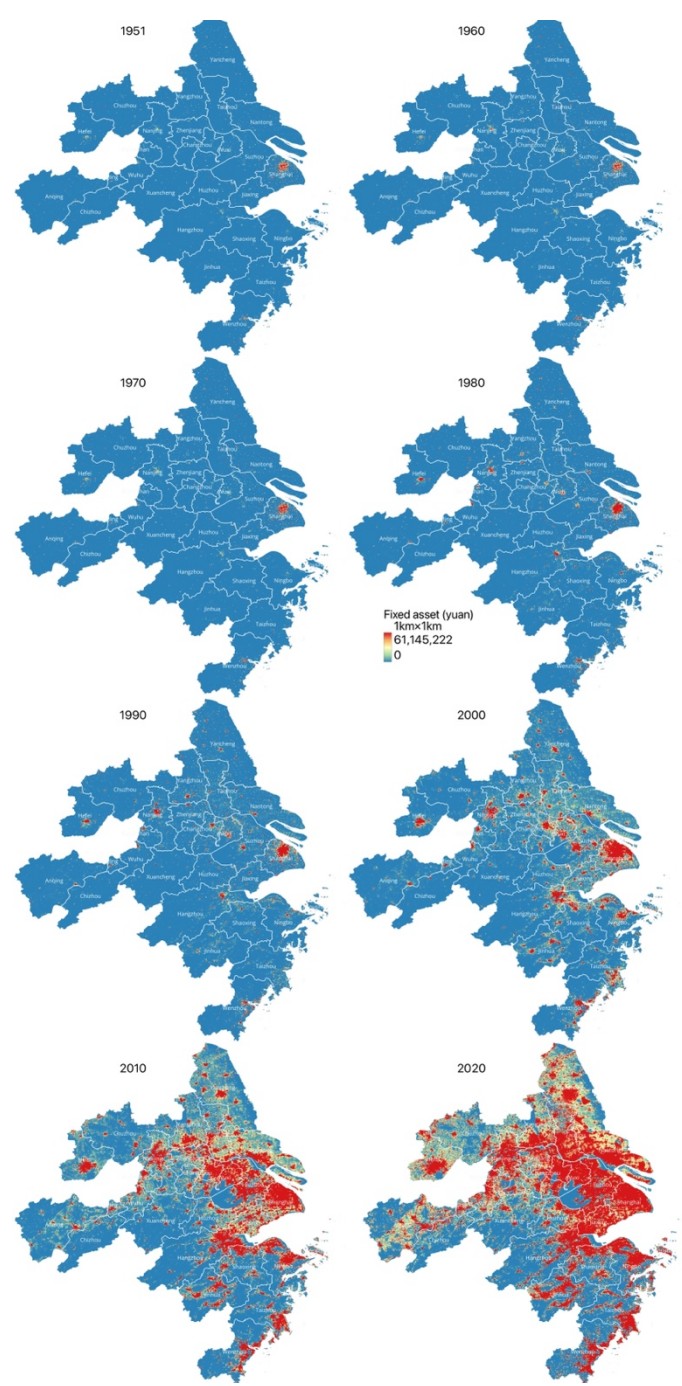

**Figure 8: The spatial distribution maps of grid-level fixed assets (adjusted to 2020 constant price level) modelled for the Yangtze River Delta Urban Agglomeration (YRD-UA) in 1951, 1960, 1970, 1980, 1990, 2000, 2010, and 2020. The legend is generated based on the value range of fixed assets in 1980 and is uniformly applied to the asset maps of other years for better visualization effect.**

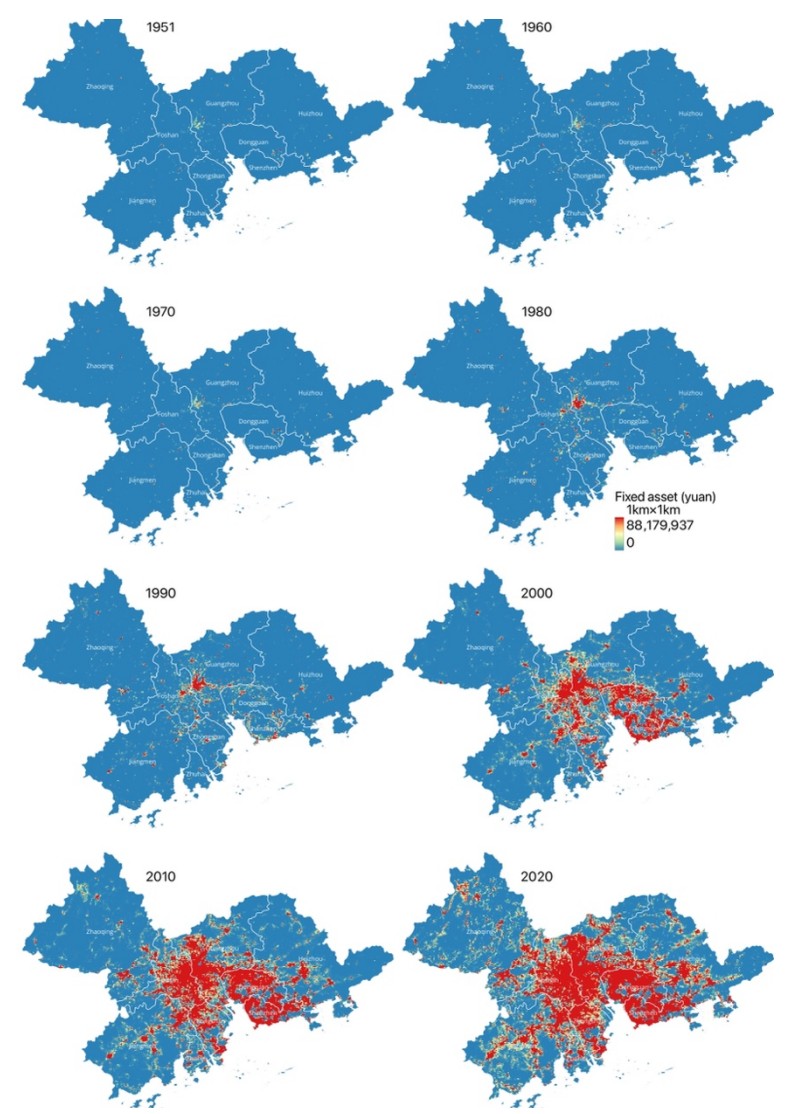

**Figure 9: The spatial distribution maps of grid-level fixed assets (adjusted to 2020 constant price level) for the Pearl River Delta Urban Agglomeration (PRD-UA) in 1951, 1960, 1970, 1980, 1990, 2000, 2010, and 2020. The legend is generated based on the value range of fixed assets in 1980 and is uniformly applied to the asset maps of other years for better visualization effect.**

**4 Discussion**

**4.1 The consistency check of disaggregation indexes**

As introduced in Section 2.2, different combinations of nighttime light, population, and built-up surface area data are employed to generate corresponding disaggregation indexes (lit-pop, area-pop, and pop-pop), considering the difference in temporal availability of these ancillary data. To evaluate the consistency of disaggregated grid-level fixed assets for three periods 470 (namely 1991-2020, 1971-1990, and 1951-1970) by using different disaggregation indexes, it is necessary to test the correlation between these disaggregation index pairs. Therefore, by taking 2010 as the test year, three types of disaggregation index images are generated, and the correlation analyses for every two indexes of lit-pop, area-pop, and pop-pop are performed for 344 prefectures in China, as plotted in Figure 10. In this figure, the ratio between the prefectural sum and the provincial sum of

each disaggregation index value is calculated for each prefecture. The high correlation between area-pop and lit-pop (with $R^2$

= 0.98, as shown in panel (a) of Figure 10) indicates that it is reasonable to use the combination of built-up surface area and

population data to disaggregate the province-level fixed assets for years before 1990 when nighttime light data are not available.

The correlation between area-pop and pop-pop is the same as that between lit-pop and pop-pop (with $R^2=0.92$ for both),

indicating the acceptability of using the squared population to disaggregate the province-level fixed assets for years before

1970 when both nighttime light and built-up surface data are unavailable.

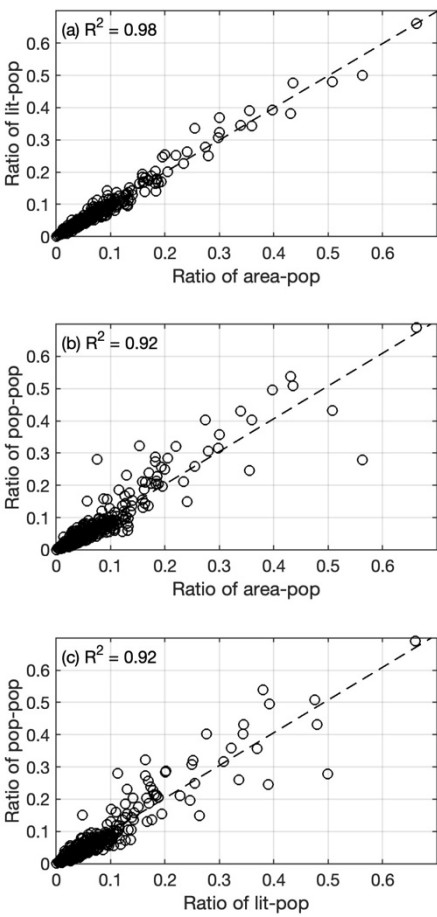

**Figure 10: The correlation between ratios of (a) area-pop and lit-pop, (b) area-pop and pop-pop, and (c) lit-pop and pop-pop for 344 prefectures in China. Note that for prefectures within each provincial-level administrative unit, the index (lit-pop, area-pop, pop-pop) ratio is derived by dividing the sum of the index value of each prefecture by the sum of the index value of the corresponding province.**

**4.2 Performance evaluation of modelled fixed assets at different scales**

Due to the lack of officially issued statistics on annually accumulated fixed assets, it is important to compare our modelled

fixed asset data with that of other studies. Wu et al. (2014) conducted a benchmark estimation of wealth capital stock in 344

prefectures of China from 1978 to 2012 using the PIM, providing both prefecture-level and provincial-level fixed asset values

for 2012. Therefore, we first compare our modelled asset values with those provided by Wu et al. (2014) for 2012 at the

provincial level, as listed in Table 3. The ratio between our modelled fixed assets and those in Wu et al. (2014) is between 0.95

to 1.93, with the largest deviation occurring in the estimation for Anhui province. According to the comparison analysis by Li

(2011), a 1% change in the depreciation rate will lead to a 10% change in accumulated fixed asset 25 years later. Therefore, the deviation in estimated fixed asset for Anhui province might be partly due to the difference in the depreciation rate used, which is 5% in this paper and 9.75% in Wu et al. (2014). Additionally, differences in the compiled implicit deflator series used to calibrate the deflation of TIFA over time may also contribute to the deviation. We also compare our modelled fixed assets for 344 prefectures in China with those by Wu et al. (2014) for 2012. As shown in Figure 11, the correlation coefficient of these two datasets at the prefecture level is relatively high (with $R^2 = 0.95$), indicating their good consistency. Similar to the reason explained for the discrepancy in Table 3, the consistently high fixed assets at the prefecture level estimated in this paper are probably due to the relatively low depreciation rate used, which is uniformly set at 5%, while in Wu et al. (2014), the depreciation ratio ranges from 7.95% to 10.05% for different provinces.

In our previous work, a grid-level residential building stock model for mainland China was developed based on urbanity-level (urban, township, and rural) population and building-related statistics in each province extracted from the records in the tabulation of the 2010 population census of China (Xin et al., 2021). Therefore, we also conduct a correlation analysis between the modelled residential building replacement values in Xin et al. (2021) (without considering depreciation) and the fixed assets modelled in this paper (including residential and non-residential buildings, infrastructures, instruments, etc., with depreciation over time considered) for all 344 prefectures. Figure 12 shows their correlation is also relatively high (with $R^2 = 0.91$). The two obvious deviation points in Figure 12 correspond to Shanghai and Beijing. The reasons for such deviations are complex and related to multiple factors, including whether depreciation is considered and discrepancies in the unit construction prices chosen for different residential buildings in Xin et al. (2021) compared with the price levels used for fixed assets in this paper, as they are determined through quite different price compilation channels.

**Table 3: Comparison of the estimated values of accumulated fixed assets (in billion Chinese yuan) in 2012 for 31 provinces of mainland China in this paper and Wu et al. (2014).**

| Province | This study | Wu et al. (2014) | Ratio |
|---|---|---|---|
| Beijing | 4649.23 | 3849.16 | 1.21 |
| Tianjin | 3946.97 | 3883.61 | 1.02 |
| Hebei | 10092.73 | 6822.05 | 1.48 |
| Shanxi | 4380.54 | 3269.85 | 1.34 |
| Inner Mongolia | 5871.68 | 5387.49 | 1.09 |
| Liaoning | 10633.95 | 6820.13 | 1.56 |
| Jilin | 4841.46 | 4516.33 | 1.07 |
| Heilongjiang | 4776.97 | 3194.81 | 1.50 |
| Shanghai | 5124.19 | 4568.20 | 1.12 |
| Jiangsu | 16962.45 | 12728.77 | 1.33 |
| Zhejiang | 10902.16 | 7797.98 | 1.40 |
| Anhui | 7436.09 | 3856.29 | 1.93 |
| Fujian | 5813.26 | 4730.33 | 1.23 |
| Jiangxi | 5444.77 | 2931.79 | 1.86 |
| Shandong | 17724.37 | 13174.73 | 1.35 |
| Henan | 10954.13 | 9302.42 | 1.18 |
| Hubei | 7482.44 | 5440.21 | 1.38 |
| Hunan | 6971.73 | 5221.56 | 1.34 |

| | | | |
|---|---|---|---|
| Guangdong | 12272.68 | 10655.79 | 1.15 |
| Guangxi | 4479.82 | 4738.49 | 0.95 |
| Hainan | 998.95 | 785.94 | 1.27 |
| Chongqing | 4546.95 | 2981.25 | 1.53 |
| Sichuan | 8854.82 | 5767.49 | 1.54 |
| Guizhou | 2475.02 | 2078.73 | 1.19 |
| Yunnan | 3963.58 | 3272.57 | 1.21 |
| Xizang | 383.77 | 337.04 | 1.14 |
| Shaanxi | 5563.17 | 4253.29 | 1.31 |
| Gansu | 2268.29 | 1560.18 | 1.45 |
| Qinghai | 849.86 | 626.12 | 1.36 |
| Ningxia | 1028.43 | 853.45 | 1.21 |
| Xinjiang | 2955.60 | 2189.56 | 1.35 |
| SUM | 164749.28 | 147595.58 | 1.12 |


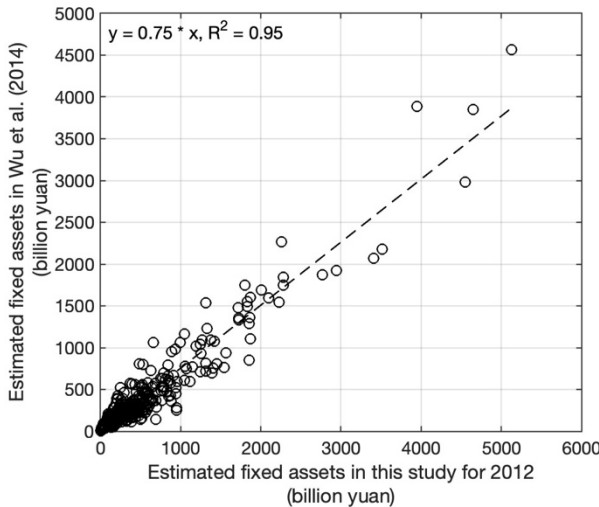

**Figure 11: The correlation analysis between the estimated fixed assets for 344 prefectures of mainland China in this paper and those given in Wu et al. (2014) for 2012.**

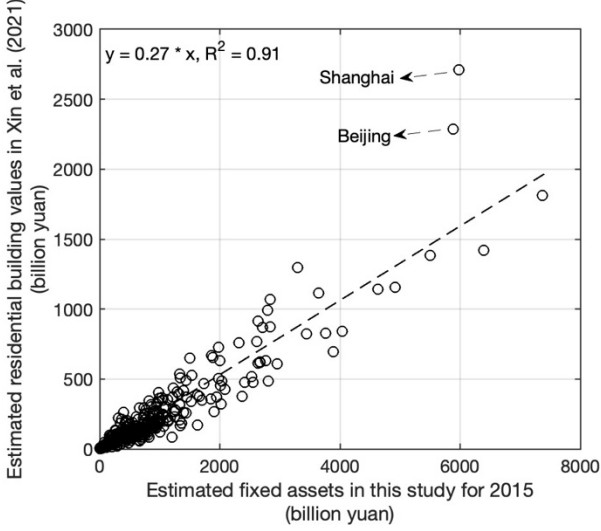

**Figure 12: The correlation analysis between the estimated fixed assets for 344 prefectures of China in this paper and the estimated residential building replacement values in Xin et al. (2021) for 2015.**

**4.3 Application of the fixed asset data to seismic loss estimation of damaging earthquakes**

Since the development of the fixed asset data is targeted for seismic loss estimation after the occurrence of a damaging earthquake in China, which is of vital reference importance for government officials to reasonably allocate emergency response

personnel and goods as well as for insurance and reinsurance companies to quickly estimate the potential compensation amount, it is necessary to check the application potential of the developed dataset. Taking the seismic loss estimation for the Ms6.2 Jishishan earthquake that occurred on December 18, 2023, in Gansu province, China, as an example, we overlap the modelled grid level fixed asset data of year 2020 in the earthquake-stricken area with the officially issued macro-seismic intensity map by China Earthquake Administration, as shown in Figure 13. The sum of exposed fixed asset value within each intensity extent

can be calculated by using open-source software QGIS. Then, the mean loss ratio within each intensity to be used for loss calculation is determined from the empirically regressed loss estimation model for the Qinghai-Tibetan region of China (a broader region that covers the study area in Figure 13) in our previous work (Li et al., 2023), in which the provincial-level fixed asset data developed in this paper, the digitalized macro-seismic intensity maps and loss records of damaging earthquakes occurred in the past seven decades in China compiled by Li et al. (2021), and the LandScan population density data (Bhaduri

et al., 2007) are comprehensively used to regress the empirical economic loss models for different sub-regions in China. By multiplying the mean loss ratio for each intensity with its accumulated fixed asset value, the direct economic loss of the Jishishan earthquake is estimated to be 10.69 billion Chinese yuan. Such a rough estimation (since no building or infrastructure attribute information is used and the newly added investment in fixed asset during 2021-2023 is not considered ) is quite comparable to the officially issued total loss of 14.6 billion Chinese yuan, which is calculated based on a detailed post-

earthquake investigation of damaged buildings and infrastructures due to the earthquake (ONDP-MRC, 2024). The difference between these two loss numbers is within two times, which indicates that our estimated loss can be considered a reasonable loss estimation result. This application to seismic loss estimation also verifies that the fixed asset data developed in this paper is promising to be used for rapid economic loss estimation for future damaging earthquakes in China.

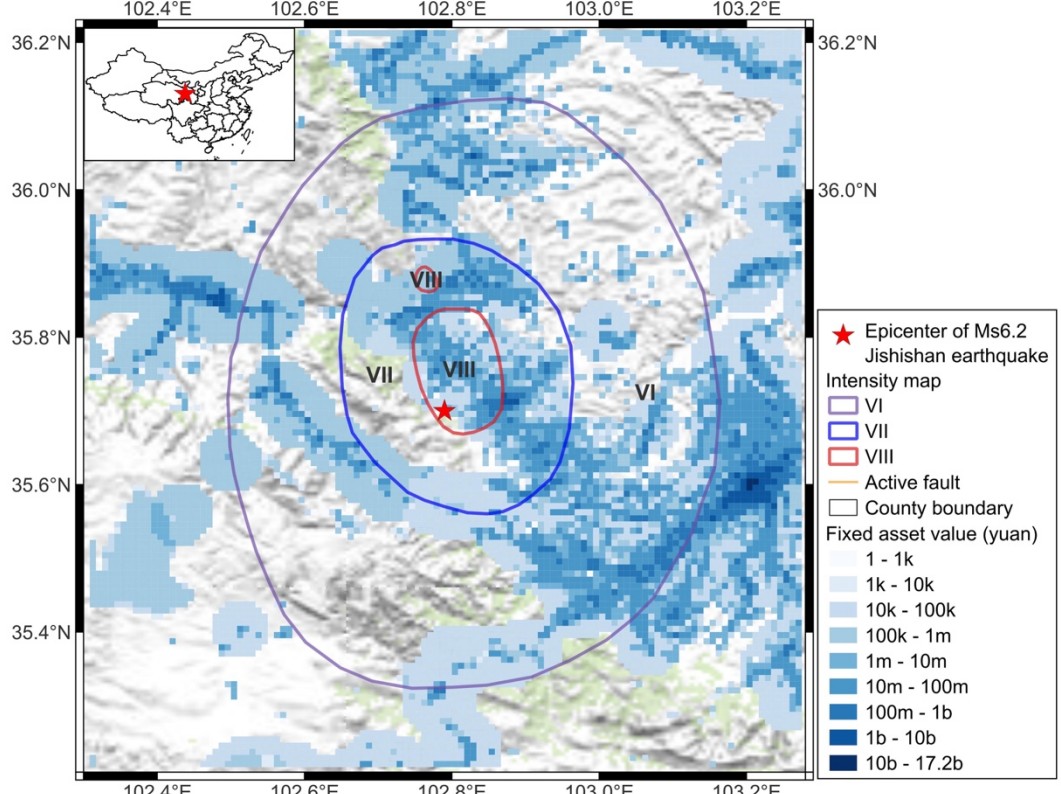

**Figure 13: Overlap the modelled fixed asset data for the year 2020 at grid level in the earthquake-stricken area with the seismic intensity map of the 2023.12.18 Ms6.2 earthquake that occurred in Gansu province, China (with its location given in the inset map). The text "k, m, b" in the legend of the fixed asset refers to thousand, million, and billion, respectively. The active fault traces are from Wu et al. (2024) and the basemap is from the "USGC Topo Mohit" map provided by the plug-in of "QuickMapServices" in QGIS.**

Like seismic risk analysis, different risk assessment scales require different methods and data inputs. It is also worthwhile to discuss the applicability of the developed fixed asset data to risk analysis of other natural hazards (e.g., flood, high/extreme wind, etc.). In this paper, the fixed asset model is developed based on the Level 1 data, which mainly include demographic data and fixed asset investment related statistics extracted from the national census and yearbooks but no structure characteristics. Therefore, the disaggregated fixed asset model based on this level of information is relatively rough. Previous

studies (Röthlisberger et al., 2018; Wouters et al., 2021; Wu et al., 2019) suggest that to get reasonable flood risk assessment results, detailed building attributes (floor area, height) should be gained to better assess building vulnerability to flood. As further verified by the comparison of different exposure models for flood risk analysis in Röthlisberger et al. (2018), the estimation of exposed building values should be based on individual buildings rather than on areas of land use types. However, from the fixed asset data developed in this paper, we cannot directly derive the floor area or height information even at the

administrative scale due to the lack of more details, let alone for individual buildings. Even at the grid level, flood loss estimation results are more sensitive to changes in exposure resolution than seismic loss estimation results. For example, the test by Dabbeek et al. (2021) revealed that when changing the resolution of the exposure model from 1km to 8km, the change in final seismic loss is less than 5%. In contrast, Bouwer et al. (2009) found that using a 100-m instead of a 25-m exposure grid in loss estimation would result in increased flood damage estimates up to 50% higher. In the case of high/extreme wind

risk analysis, studies published in recent years also tend to use building level or even component level exposure data to assess wind risk (Pandolfi et al., 2022). Such component level building attributes cannot be derived from our current fixed asset model either. Therefore, we consider the fixed asset data developed in this paper is more suitable for rapid seismic loss estimation after the occurrence of damaging earthquakes in China and should be used with caution when applying it to the risk analysis of other types of natural hazards.

**4.4 Limitations of the modelled fixed asset data**

Inevitable assumptions and simplifications have been made during the modelling process of the fixed asset data. For example, a fixed depreciation rate of 5% is uniformly applied to all provinces and periods regardless of their potential temporal and spatial differences in the depreciation rates of fixed asset types. This simplified approach may lead to skewed asset values, particularly in provinces with unique economic trajectories or asset compositions. For instance, in industrialized regions (such

as the old industrial canters of Heilongjiang, Liaoning, and Jilin provinces in China), assets may have a shorter service life than in less industrialized provinces, affecting the accuracy of economic loss projections. Therefore, it is quite necessary to integrate the temporal and spatial changes in depreciation rates when modelling the net value of depreciated fixed assets, should the statistics data required to differentiate such rates be accessible for the period 1951-2020 considered in this paper. As a matter of fact, in the prefecture-level fixed asset modelling work of Wu et al. (2014) for China during 1978-2012, they

did develop varying depreciation rates for different provinces (see their Table 2). To get these depreciation rates, the residual value of the capital stock was uniformly set to be 4% of their original value, and the service lives of three different fixed asset types (construction and installation, equipment and instruments, and others) in all provinces were uniformly set as 45 years, 20 years, and 25 years, respectively. Then, following the expression like Eq. (3) in this paper, the national-average depreciation rate for each asset type was determined accordingly by Wu et al. (2014), which was 6.9%, 14.9%, and 12.1%, respectively.

Considering the composition of fixed asset may vary across provinces, to better reflect their impact on spatial differences in depreciation rates among provinces, Wu et al. (2014) further multiplied the relative share of each fixed asset type in each province (such data are available from 1983) with its depreciate rate derived above. Then, the average depreciation rate can be determined for each province. For cross-check convenience, these provincial-level depreciation rates developed by Wu et al. (2014) are also provided in Table 4 of this paper.

**Table 4: The provincial-level depreciation rates modified from Table 2 of Wu et al. (2014).**

| Province | Depreciation rate (%) |
|---|---|
| Beijing | 9.76 |
| Tianjin | 9.56 |
| Hebei | 9.83 |
| Shanxi | 9.53 |
| Inner Mongolia | 9.22 |
| Liaoning | 9.49 |
| Jilin | 9.43 |
| Heilongjiang | 9.28 |
| Shanghai | 10.05 |

| | |
|---|---|
| Jiangsu | 9.62 |
| Zhejiang | 9.53 |
| Anhui | 9.75 |
| Fujian | 9.44 |
| Jiangxi | 9.68 |
| Shandong | 9.49 |
| Henan | 9.45 |
| Hubei | 9.58 |
| Hunan | 9.20 |
| Guangdong | 9.35 |
| Guangxi | 9.44 |
| Hainan | 9.35 |
| Chongqing | 9.21 |
| Sichuan | 9.27 |
| Guizhou | 9.35 |
| Yunnan | 9.03 |
| Tibet | 7.95 |
| Shaanxi | 9.87 |
| Gansu | 9.30 |
| Qinghai | 8.74 |
| Ningxia | 9.32 |
| Xinjiang | 9.08 |
| **Average rate** | 9.39 |

Note: the "Average rate" of 9.39% in the last row of this table is simply the averaged value of all these 31 provincial level depreciation rates.

However, although the regional difference in fixed asset composition is considered when developing the averaged depreciation rate for each province, the depreciation rates listed in Table 4 for most provinces are around 9%, ranging from 595    7.95% to 10.05%. This means the spatial difference in depreciation rate among provinces is basically within ±1%, which is smaller than being imagined. And 1% change in depreciation rate would lead to no more than 10% change in accumulated fixed asset 25 years later, as revealed by Li (2011). They also suggested that the depreciation rate should be within the range of 5%~10%. When we further apply the depreciation rates in Table 4 to model the fixed assets at the provincial level following the process in Section 2.1, the ratios between these newly modelled fixed assets and those modelled by using a 600    uniform dereplication rate of 5% are plotted in Figure 14.

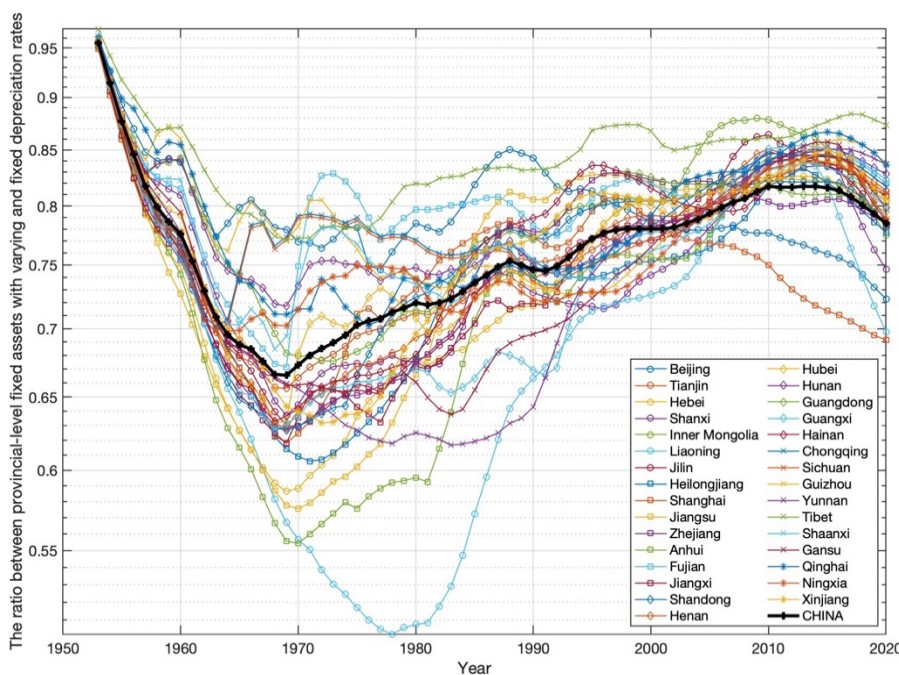

**Figure 14: The ratio between fixed assets modelled by using varying depreciation rates in Table 4 and the fixed dereplication rate of 5% for 31 provinces and mainland China as a whole.**

Figure 14 indicates that the changing trend in fixed asset ratio with time varies across provinces. But a general trend for most provinces is that the fixed asset ratio keeps decreasing before 1970 and turns to increase after 1970. The decreasing trend before 1970 is considered mainly dominant by using larger depreciation rates than the fixed rate of 5%. The increasing trend in fixed asset ratio after 1970 is related to the increase in annual investment in fixed capital stock, which overturns the decreasing trend caused by using larger depreciation rates. The changing trends of fixed asset ratios in Figure 14 also imply that the modelling of accumulated fixed asset is affected by a combination of multiple factors with varying significance in different periods. Considering the fixed asset data modelled in this paper is mainly served for seismic loss estimation and rapid emergency response after the occurrence of damaging earthquakes in China, and the values of different types of fixed asset in earthquake-affected areas are generally higher than their residual values even they have lasted for a much longer time than their service lives; therefore, a conservative depreciation rate of 5% is preferred for all provinces to get the final accumulated fixed asset data series from 1951 to 2020.

For evaluation of the grid-level fixed asset model disaggregated from provincial-level data in the early periods, a direct comparative analysis with related statistical records would be quite valuable. However, such comparison is unfortunately hindered by the lack of prefecture/county level statistical records of GDP or industrial output data in the early periods, as can be checked from the official website of the National Bureau of Statistics (https://data.stats.gov.cn/english/). Therefore, considering the rough estimation made in the determination process of the initially accumulated fixed assets as well as the lack of an official and standard method in the compilation of economic indicators in the early periods after 1949, special attention should be paid when applicating the developed grid-level fixed asset data to studies like long-term economic trend analyses for specific regions.

It is also noteworthy that in the fixed asset data modelled in this paper, these is no further differentiation of structural specificity, which limits the model's utility for applications that require asset type differentiation, such as insurance underwriting or infrastructure resilience planning. It could lead to misaligned resource allocations during emergency responses under certain circumstances. For studies devoted to modelling the fixed asset values for specific years (e.g., Gunasekera et al., 2015; Wu et al., 2018; Xin et al., 2021), it is possible to use more detailed census records or even in-site investigation data to estimate the value of different asset types. However, when accumulated fixed asset data series for an extended period dating back to 1951 are needed, as in this paper, detailed statistics to differentiate the building types (residential/industrial/commercial) or even the quota of different fixed asset types (buildings, infrastructures, instruments) exposed to past years are typically missing. In this case, the fixed asset model can only be developed from the Level 1 data (as explained in the Introduction section). Considering the completeness and availability of statistical data during 1951-2020, the perpetual inventory method (PIM) is a more appropriate choice to develop the provincial level fixed asset data series. Although there is no differentiation of building types in the fixed asset data developed in this paper, when compared the modelled asset value with the estimated replacement value of residential buildings (including 17 building sub-types) in our previous work for China (Xin et al., 2021), as shown in Figure

12, we find the correlation at prefecture level between these two models remains relatively high (with $R^2$=0.91). In the future, to better aid the seismic risk assessment and emergency response needs, more efforts should be made to employ more detailed building and infrastructure related statistics to refine the latest fixed asset model. For example, land-use data can be integrated into the modelling process to better discriminate land types and exposed elements, detailed building-related census records can

be adopted to better quantify the asset share of different building types, and road density data can be used to well improve the asset evaluation and disaggregation accuracy of infrastructures.

## 5 Code and data availability

The modelled provincial-level fixed asset data for 31 provincial administrative units in China from 1951 to 2020 has been up loaded to Zenodo (https://doi.org/10.5281/zenodo.12706096) (Xin et al., 2024). The nighttime light data from 1992 to 2000 w

ith a spatial resolution of 30 arc-seconds compiled by Li et al. (2020) are available from Figshare (https://figshare.com/article s/dataset/Harmonization_of_DMSP_and_VIIRS_nighttime_light_data_from_1992-2018_at_the_global_scale/9828827). The population and built-up surface datasets used in this paper are provided by the Global Human Settlement Layer (GHSL) proj ect of the Joint Research Centre, European Commission (https://human-settlement.emergency.copernicus.eu/datasets.php).

## 6    Conclusions

This paper develops grid-level fixed asset data for China from 1951 to 2020 based on the perpetual inventory method (PIM) and disaggregation techniques, aiming to improve the accuracy of seismic loss estimation for damaging earthquakes in China. Consistency checks have demonstrated the model's reasonableness and reliability. However, the fixed asset values are derived from investment related data recorded in statistical yearbooks, and detailed information on building structures and infrastructures is not included in the modelling process. Therefore, the datasets are primarily intended to facilitate rapid

estimation of seismic losses, serving as a crucial reference for the government in formulating emergency response plans and allocating rescue personnel and resources shortly after the occurrence of damaging earthquakes. To demonstrate the application of the developed fixed asset data in seismic loss estimation, the estimated loss in this paper for the 2023 Ms6.2 Jishishan earthquake is compared with the officially issued loss. Their consistency further verifies the applicability potential of the developed fixed asset data to seismic loss estimation for future damaging earthquakes in China. The modelled fixed asset data

from 1951 to 2020 can be conveniently extended to more recent years as new fixed asset-related statistics become available.

## Author Contributions

DX designed the approach, collected the data, performed the analyses, and prepared the draft manuscript. JD and FW gave

quite constructive suggestions when preparing the manuscript. ZZ guided the project and provided the financial support. ZZ, SW, and XF proposed very inspiring comments and valuable application suggestions. All authors contributed to the revision of the manuscript.

**Competing interests**

The authors declare that there are no competing interests.

**Acknowledgements**

The authors deeply thank the editor Solmaz Mohadjer for actively monitoring the whole review process, carefully checking the technical shortcomings and timely responses to our requests. We also sincerely appreciate the efforts and time spent by Prof. Zoran Stojadinovic and another anonymous reviewer for their constructive comments and suggestions, which have greatly improved the quality of this work.

**Financial support**

This work is supported by the National Natural Science Foundation of China (Grant Number 42204054), the National Key R&D Program of China (2023YFC3008604), Shenzhen Science and Technology Program (Grant Number JCYJ20241202125759001), and the Shenzhen Stable Support Plan Program for Higher Education Institutions (Grant Number 20220815084720001).

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
