# Peer review of "The grid-level fixed asset model developed for China from 1951 to 2020"

_Natural Hazards and Earth System Sciences, 2024_

## Author Comment (AC1)

**RC1 comments and responses:**

The manuscript outlines a methodology for constructing a time-dependent fixed asset model applicable to China. It establishes a model using fixed asset data at the provincial level and further refines it to a more detailed grid level with remote sensing ancillary data. The findings are validated with results from other studies. The methodology is purported to assist in quick post-earthquake loss estimations. The topic is interesting, and the manuscript is well-written. However, it can benefit from some comments I suggest below to improve its clarity and contribution to the field.

**Response:** Thank you very much for your time and efforts spent on reviewing our manuscript, which is deeply appreciated. And we are really touched by your careful check of the whole manuscript! We have read all your constructive comments and suggestions carefully and tried our best to prepare the initial responses. Hopefully, they can help release the concerns you have on the current version of the manuscript. Our detailed responses are given as follows.

**General comment:**

The methodology and results seem useful for assessing economic losses due to any hazard, not only due to earthquakes. Even though the introduction includes references about the importance of earthquakes and previous earthquake loss estimations, the methodology and results do not confront loss estimations from past earthquake events. Furthermore, Fig. 2 has no input that one could identify with an earthquake hazard or risk (e.g., ground motion fields, earthquake occurrence rate, or fragility model). The methodology and results focus on modelling the observed economic and demographic growth in the study areas consistently, considering the data quality limitations in terms of spatial resolution and time-frequency. I think the paper can be stronger (and closer to the journal's scope) by adding some statements in the discussion and conclusion about how the methodology and results can help in the loss estimation due to an earthquake event or other natural hazards. It would also be interesting to add to the introduction a review of studies of other natural hazards, such as floods or extreme wind events.

**Response:** Thank you for further recognizing the applicability of this fixed asset dataset to the risk assessment of other hazards. As to its application to seismic loss estimation, we have an independent paper (Li et al., 2023) demonstrating in detail how this fixed asset data can be combined with intensity maps and damage information of historical earthquakes to regress empirical loss models for different regions in China. In this paper, examples were also provided to demonstrate the loss estimation for recently occurred damaging earthquakes. Therefore, in this paper we limit ourselves to focus on the introduction, disaggregation, and evaluation of this fixed asset dataset. This explanation will be added to the revised manuscript.

Like seismic risk analysis, different scales call for different methods when it comes to assessing flood/wind risks and assessments at different scales have different uses. As explained in Page 2, Lines 34-46 in the manuscript, the fixed asset model developed in this paper is based on the Level 1 data. Based on this level information, the estimated seismic loss is relatively a rough estimation since the input data mainly include demographic data and building-related statistics extracted from the national census. When applicating the developed fixed asset data to other natural hazards (e.g., flood or extreme wind event), the required resolution and attributes of exposed asset will be different.

For flood risk assessment, detailed building attributes (floor area, height) value should be gained to better assess building vulnerability to flood (Röthlisberger et al., 2018; Wouters et al., 2021; Wu et al., 2019). As verified by the comparison of different exposure models for floor risk analysis in Röthlisberger et al. (2018), the estimation of exposed building values should be based on individual buildings rather than on areas of land use types. However, from the fixed asset data developed in this paper, we cannot further derive the floor area and height information even for administrative units, let alone for individual buildings. Therefore, we consider the developed fixed asset data may not be applicable enough for reliable flood risk analysis. Even at grid level, the flood loss estimation results are more sensitive to changes in exposure resolution than seismic loss estimation results. For example, the work of Dabbeek et al. (2021) revealed that when changing the resolution of exposure model from 1km to 8km, the change in final overall loss is less than 5%. In contrast, the work of Bouwer et al. (2009) found that using a 100-m grid instead of a 25-m grid for the same case study area resulted in damage estimates up to 50% higher. In the case of wind risk analysis, studies published in recent years also tend to use building level or even component level exposure data to assess the wind risk (Pandolfi et al., 2022). Such building attributes cannot be derived from our current fixed asset model either. This comparison will be added to the revised manuscript.

Therefore, we consider targeting the application of the developed fixed asset data to seismic loss estimation, for which reasonable estimation performance have been achieved as demonstrated in our another work (Li et al., 2023). In addition, there are already quite a few review articles written by specialists focusing on risk analysis of flood and wind. Among them, De Moel et al. (2015) provided a quite comprehensive overview on the current state, development, assessment characteristics of quantitative flood risk assessments at different scales (supra-national, macro, meso, micro). They also outlined the lessons learnt from current practice and identified future research needs for flood risk assessment. Yu et al. (2023) provides a comprehensive review on the studies related to exposure roughness

exposed to wind hazard. Therefore, we will recommend readers to refer to their work for more comprehensive information in the revised manuscript.

**Specific comment:**

1. The conclusion says the methodology can be extended to more recent years once the data is available. However, considering that the methodology aims to help in quick loss assessment for future earthquake events, can the methodology with the available data today (in 2024) provide a prediction, for example, of the losses after an earthquake event in 2030?

   **Response:** Thank you for this question. The direct answer is yes. Since the estimation of seismic loss is based on the combination of intensity map, fixed assets affected by the earthquake, and the vulnerability curve of fixed assets, for an earthquake to occur in 2030, we would recommend the use of latest fixed asset map available to assess its seismic loss, as long as the intensity map can be reasonably modelled, and the increased fixed assets from 2024 to 2030 can be added to the fixed asset map for 2030.

2. I suggest mentioning the future availability of grid-level fixed asset data only in the section "code and data availability", as it is done and justified in this version of the manuscript, and only mentioning it in the abstract and conclusion in a later version, when the data is effectively available.

   **Response:** Thank you for this suggestion. We will remove the expression related to data availability in the abstract and conclusion in the revised manuscript.

**Technical corrections**

1. Tables 2 and 3, as well as Figs. 5, 7-9: Consider changing the monetary units to billion Chinese yuans, as done in Figs. 11-12

   **Response:** Thank you for this suggestion. In Tables 2 and 3, the monetary units will be changed to billion Chinese yuan in the revised manuscript. However, in Figs. 5.,7-9, to better demonstrate the spatial locations of clusters of high fixed assets, the upper threshold of fixed asset value in the color bar is set as 243801864, 74247334, 61145222, and 88179937 in Fig. 5, Fig. 7, Fig. 8, and Fig. 9, respectively, which corresponds to the 98% quantile of the grid-level fixed asset value in each figure. These values are all smaller than 1 billion since these values only represent fixed asset in the 1km×1km grids. When the unit in these figures is expressed in billion Chinese yuan, the numbers at grid level will be too small to differentiate the spatial clusters of high fixed assets. Therefore, for better visualization effect, the monetary unit of yuan is a relatively better choice.

2. Fig. 2: There is a typo in one of the charts: "Harmonized" instead of "Harmanized".

   **Response:** Thank you so much for your careful check! This error will be rectified in the revised manuscript.

3. Although described in the text, the delta in Eq. 3 has a different meaning than the delta in Eq. 4. I suggest using a different symbol for one of them.

   **Response:** Thank you so much for this suggestion! We will use $\kappa$ to replace $\delta$ in Eq. (3) in the revised manuscript.

4. Table 1: There is a typo in the 2nd column, 8th row: "Population density data" instead of "Population dentsity data".

   **Response:** Thank you again for your so careful check! This typo will be rectified in the revised manuscript.

**References mentioned in the responses:**

Bouwer, L. M., Bubeck, P., Wagtendonk, A. J., and Aerts, J. C. J. H.: Inundation scenarios for flood damage evaluation in polder areas, Nat. Hazards Earth Syst. Sci., 9, 1995–2007, https://doi.org/10.5194/nhess-9-1995-2009, 2009.

Dabbeek, J., Crowley, H., Silva, V., Weatherill, G., Paul, N., and Nievas, C. I.: Impact of exposure spatial resolution on seismic loss estimates in regional portfolios, Bull Earthquake Eng, https://doi.org/10.1007/s10518-021-01194-x, 2021.

De Moel, H., Jongman, B., Kreibich, H., Merz, B., Penning-Rowsell, E., and Ward, P. J.: Flood risk assessments at different spatial scales, Mitig Adapt Strateg Glob Change, 20, 865–890, https://doi.org/10.1007/s11027-015-9654-z, 2015.

Li, B.: Comparative Analysis of Estimates on Capital Stock of China, Journal of Quantitative and Technological Economics, 21–54, https://doi.org/10.13653/j.cnki.jqte.2011.12.006, 2011.

Li, Y., Xin, D., and Zhang, Z.: Estimating the economic loss caused by earthquake in Mainland China, International Journal of Disaster Risk Reduction, 95, 103708, https://doi.org/10.1016/j.ijdrr.2023.103708, 2023.

Pandolfi, F., Baltzopoulos, G., and Iervolino, I.: ERMESS: extreme wind risk assessment for building portfolios, Nat Hazards, https://doi.org/10.1007/s11069-022-05740-x, 2022.

Röthlisberger, V., Zischg, A. P., and Keiler, M.: A comparison of building value models for flood risk analysis, Nat. Hazards Earth Syst. Sci., 18, 2431–2453, https://doi.org/10.5194/nhess-18-2431-2018, 2018.

Wouters, L., Couasnon, A., de Ruiter, M. C., van den Homberg, M. J. C., Teklesadik, A., and de Moel, H.: Improving flood damage assessments in data-scarce areas by retrieval of building characteristics through UAV image segmentation and machine learning – a case study of the 2019 floods in southern Malawi, Nat. Hazards Earth Syst. Sci., 21, 3199–3218, https://doi.org/10.5194/nhess-21-3199-2021, 2021.

Wu, J., Ye, M., Wang, X., and Koks, E.: Building Asset Value Mapping in Support of Flood Risk Assessments: A Case Study of Shanghai, China, Sustainability, 11, 971, https://doi.org/10.3390/su11040971, 2019.

Yu, J., Stathopoulos, T., and Li, M.: Exposure factors and their specifications in current wind codes and standards, Journal of Building Engineering, 76, 107207, https://doi.org/10.1016/j.jobe.2023.107207, 2023.

---

## Author Comment (AC2)

**RC2 comments and responses**

**The overall quality of the preprint (general comments)**

The overall quality of the paper is high. The topic of developing a novel fixed asset model to improve seismic loss estimation is significant for the science community. By mapping fixed assets at a 1 km × 1 km grid level, the model better serves rapid seismic loss assessments and informs emergency response plans. The research is well structured and explained. The authors made an effort to combine various data sources and techniques. While the model represents a significant advancement, several limitations could impact its effectiveness, particularly in high-stakes applications like earthquake response. The paper's scientific contributions justify publication with minor revisions to handle specific data assumptions better and further validate the model's application. These adjustments would help ensure the model's broader applicability and robustness.

**Response:** We deeply appreciate the time and efforts you have denoted to improving the quality of this manuscript and thank you so much for all the constructive comments! Our initial responses are given as follows. Hopefully they can help release your concern on the current version of the manuscript.

**Individual scientific questions/issues (specific comments)**

Here are some topics which the authors could discuss in more detail:

**Reliance on Historical Investment Data and Simplified Depreciation Rates**. The model bases its estimates on historical investment data and applies a uniform 5% depreciation rate across all provinces, regardless of variations in asset type, economic condition, or regional maintenance practices. The uniform deprecation rate can introduce inaccuracies, especially for assets with different service lives or conditions. The simplified approach to depreciation may lead to skewed asset values, particularly in provinces with unique economic trajectories or asset compositions. For instance, in industrialized regions, assets may have a shorter useful life than in less industrialized provinces, affecting the accuracy of economic loss projections. Can the model be refined by including a variable depreciation rate based on more detailed asset-specific and regional data, if available?

**Response:** Thank you for this comment and we totally agree with your improvement suggestions. And it would be possible to refine the developed fixed asset model by including variable depreciation rates with its temporal and spatial change being considered, should the statistical data required to determine such rates be accessible for the period 1951-2020.

As a matter of fact, in a prior prefecture-level fixed asset modeling work of Wu et al. (2014) for China during 1978-2012, they did develop different depreciation rates for each province, as given in the following Table R1. To derive these depreciation rates, the residual value

of the capital stock was set to be 4% of their original value, and the service life of different fixed asset types (construction and installation, equipment and instruments, and others) was set as 45 years, 20 years, and 25 years, respectively. Following Eq. (3) of the manuscript, the national-average depreciation rate can be calculated accordingly as 6.9%, 14.9%, and 12.1%, respectively. To better reflect the spatial difference in depreciation rate among provinces, Wu et al. (2014) further used the relative share of each fixed asset type in each province published in provincial statistical yearbooks dating back to 1983. Then for each province, the depreciation rate for each province can be determined separately considering the average weight of each fixed asset, as listed in Table R1. Additionally, for each province, the deprecation rate range listed in the brackets of Table R1 was determined by changing the residual value ratio from 4% to 3% and 5%.

As can be seen from Table R1, the depreciation rate for most provinces is around 9%, although the relative share of different fixed asset types has been considered for each province. And when the residual value ratio is 4%, the total range of depreciation rates for all provinces is from 7.95% to 10.05%. However, due to the big difference in development stage of China before 1980 and after 1980, we cannot directly apply the provincial level depreciation rates developed in Wu et al. (2014) by using statistics during 1983-2012 to years before 1980s. That's why a fixed depreciate rate of 5% was finally used in this paper, which may appear to be conservative and the reason is explained in Page 9, Lines 239-243 of the manuscript that "*since the development of provincial-level fixed capital stock data in this paper is to be used for the development of empirical loss models for rapid emergency response after the occurrence of damaging earthquakes in China, the replacement values of different types of fixed capital stock in earthquake-affected areas are generally higher than their residual values even they have lasted for a much longer time than their service lives; therefore, a conservative depreciation rate of 5% is chosen for all provinces to get the final accumulated fixed asset data series from 1951 to 2020*". The sensitivity analysis of Li (2011) indicated that the change of 1% in depreciation rate would lead to no more than 10% change in accumulated capital stock 25 years later and Li (2011) also suggested that the depreciation rate should be within the range of 5%~10%. In the performance evaluation section, when compared the modelled fixed assets with that in Wu et al. (2014) at prefecture level (as implied in Figure 11), the good correlation ($R^2$=0.95) between these two datasets help gain our confidence in the reasonability in our model. The above discussion will be added to the Discussion section of the revised manuscript.

**Table 2**
Depreciation rates across 31 provinces in Mainland China.

| Province | Depreciation rate (%) |
|---|---|
| Beijing | 9.76 (9.12, 10.58) |
| Tianjin | 9.56 (8.93, 10.36) |
| Hebei | 9.83 (9.18, 10.65) |
| Shanxi | 9.53 (8.91, 10.33) |
| Inner Mongolia | 9.22 (8.62, 10.00) |
| Liaoning | 9.49 (8.86, 10.28) |
| Jilin | 9.43 (8.81, 10.23) |
| Heilongjiang | 9.28 (8.67, 10.06) |
| Shanghai | 10.05 (9.39, 10.89) |
| Jiangsu | 9.62 (8.98, 10.42) |
| Zhejiang | 9.53 (8.90, 10.33) |
| Anhui | 9.75 (9.11, 10.57) |
| Fujian | 9.44 (8.82, 10.24) |
| Jiangxi | 9.68 (9.04, 10.49) |
| Shandong | 9.49 (8.87, 10.29) |
| Henan | 9.45 (8.83, 10.24) |
| Hubei | 9.58 (8.95, 10.38) |
| Hunan | 9.20 (8.60, 9.98) |
| Guangdong | 9.35 (8.74, 10.14) |
| Guangxi | 9.44 (8.82, 10.24) |
| Hainan | 9.35 (8.74, 10.14) |
| Chongqing | 9.21 (8.61, 9.99) |
| Sichuan | 9.27 (8.66, 10.05) |
| Guizhou | 9.35 (8.73, 10.13) |
| Yunnan | 9.03 (8.44, 9.79) |
| Xizang | 7.95 (7.42, 8.63) |
| Shaanxi | 9.87 (9.22, 10.70) |
| Gansu | 9.30 (8.69, 10.08) |
| Qinghai | 8.74 (8.17, 9.48) |
| Ningxia | 9.32 (8.71, 10.11) |
| Xinjiang | 9.08 (8.48, 9.84) |

Note: Data source: Depreciation rates are calculated by the authors, the depreciation rate inside of the parentheses is calculated by the average relative efficiency rate of 3% and 5%, respectively, as described in the text.

Table R1: The provincial level depreciation rates estimated in Wu et al. (2014).

**Inconsistencies in the Data Sources for Ancillary Datasets.** The model relies on ancillary datasets (e.g., nighttime lights, population, built-up areas) which are not consistently available across all years. This results in the use of alternative data types to approximate missing data. For instance, population data alone is used in the early years when nighttime light data is unavailable. These proxies may not accurately represent economic activity, especially in rural areas or less-developed regions, leading to potential over- or under-estimations in asset distribution. Could the model be strengthened by incorporating more recent, high-resolution satellite data or by exploring alternative disaggregation methods that do not depend solely on proxies like nighttime lights?

**Response:** Thank you for this comment. A big effort made in this paper is to find reasonable combination of ancillary data to disaggregate provincial level fixed assets into grid level. For periods (1991-2020) when nighttime light data are available, the combination of nighttime light and population are used to create the lit-pop index, which is exactly to better avoid the over- or under-estimation problems in asset distribution by using nighttime light or population data alone. For years before 1991 (1971-1990), when nighttime data are unavailable, the built-up surface area data and population data are used to create areapop index. However, for earlier periods (1951-1970) when only grid level population density data are available, we choose to apply the pop-pop index (derived from the squared value of population in each 1km×1km grid) to further disaggregate the asset value of years before 1970. And the correlation analysis in Figure 10 between each pair of three disaggregation indexes (lit-pop, area-pop, pop-pop) further validates the consistency and reasonability of these indexes.

But we do want to emphasize that for earlier period (1951-1970) when only population density data are available, we consider it is still not appropriate to incorporate the high-resolution satellite data in recent years with population data to disaggregate the asset value during 1951-1970. The reason is still that China has experienced quite different economic development stages before the 1970s and after the 1970s divided by the issuing of opening and reform policy in 1978, while before 1970 the economy development in China was very slow due to natural disaster, political movement and the planned economic system.

**Lack of Structural Detail in Asset Composition.** The model's focus on fixed capital should differentiate between asset types (e.g., residential vs. industrial buildings) in disaggregation. This lack of structural specificity reduces the model's utility for applications that require asset type differentiation, such as insurance underwriting or infrastructure resilience planning. Different asset types respond differently to seismic events; for instance, infrastructure like bridges and roads may sustain different levels of damage compared to residential buildings. This generalization could lead to misaligned resource allocations during emergency responses. Introducing asset type categorization, possibly by incorporating land use or building inventory data, would enhance the model's accuracy for specific asset loss estimations.

**Response:** We totally agree with this comment. However, when the perpetual inventory method (PIM) is used to systematically estimate the accumulated fixed asset data series for periods dating back to 1951, it means that we have no better or more detailed statistics to constrain the building types (residential/industrial/commercial) and even the quota of different fixed assets (buildings, infrastructures, instruments) exposed to past years.

As explained in Page 2, Lines 34-46 in the manuscript, the fixed asset model developed in this paper is based on the Level 1 data. Based on this level information, the estimated seismic loss is relatively a rough estimation since the input data mainly include demographic data and building-related statistics extracted from the national census. And when seismic loss is rapidly estimated after the occurrence of a damaging earthquake (for which we develop this fixed asset dataset), the vulnerability function to be used is also a quite empirical one, similar to those developed in Jaiswal and Wald (2011) in the PAGER

project. Such empirical vulnerability curve only describes the relation between general loss ratio and the macro intensity, which is regressed from damaging statistics (intensity map, total loss, exposed asset value) of historical earthquakes for specific countries and regions. And such loss estimation is quite different from the one based on the structure type and apply the corresponding vulnerability curves for specific buildings. More details are given in Page 2, Lines 48-59 of the manuscript.

In the fixed asset dataset for periods 1951-1982, we even have no more clue to confidently differentiate the quota of different fixed asset types (construction and installation, equipment and instruments, and others), which makes it even harder to differentiate different building types, as verified by the work of (Wu et al., 2014). But for studies mainly focusing on developing the replacement value for existing buildings and structures, it is possible to make such differentiation. We also made such an attempt in our previous work (Xin et al., 2021) and we also compared the modelled fixed asset value in this paper with that for residential buildings in Xin et al. (2021), as shown in Figure 12 of the manuscript. However, models with detailed building attributes for specific year is not enough for us to develop the empirical vulnerability curve, which entails the fixed asset for past years as well, as explained in Page 3, Lines 67-77 in the manuscript. Considering the availability of input data, the perpetual inventory method (PIM) is the best practice to follow to develop such asset data series.

**Technical corrections**

The article's language quality is overall sound, with a few areas where readability and formality can be improved. Here are specific suggestions regarding grammar, spelling, and phrasing.

- Maintain past tense in descriptions of the completed study. For example, in the sentence, "The fixed asset model to be developed in this paper is also based on the Level 1 data."
  **Response:** Thank you for pointing this out. We will thoroughly check the use of verb tenses when revising the manuscript. However, we have to confess that sometimes it is kind of confusing to decide which tense should be used. For example, the expression "*The fixed asset model to be developed in this paper is also based on the Level 1 data*" can also be regarded as a description of the fact. In this case, it seems the present tense should be used…

- Remove redundant phrases. For example, "To summarize, the nighttime light data and GHS-POP data are used to generate the lit-pop disaggregation indexes from 1991 to 2020, while the GHS-BUILT-S data and GHS-POP data are used to construct area-pop disaggregation indexes from 1971 to 1990..."

**Response:** Thank you for this suggestion. If our understanding is correct, do you recommend the cited expression "*To summarize, the nighttime light data and GHS-POP data are used to generate the lit-pop disaggregation indexes from 1991 to 2020, while the GHS-BUILT-S data and GHS-POP data are used to construct area-pop disaggregation indexes from 1971 to 1990...*" should be removed as a whole? The initial consideration to give such summarization in the Results section is mainly to help readers who only roughly scan this work to quickly understand the difference in ancillary data used for different periods. And the feedback given from other readers indicates it is still necessary to keep it as a reminder.

- Avoid Informal Language. For example, replacing "Luckily" with "Fortunately" is more formal.

   **Response:** Thank you for this suggestion, accepted! We have reread the manuscript carefully and replaced the expression like "Luckily" to "Fortunately" accordingly. However, we're afraid we could not thoroughly rectify or recognize all the remaining informal expressions due to the long-time usage habit… But if the manuscript can be fortunately accepted, the NHESS journal will also offer excellent language editing service before publishing the final version of the manuscript, which will help ensure that all the expressions will be formal.

- Revise for Consistency in Abbreviations and Acronyms. Introduce abbreviations consistently upon first mention, ensuring they are used uniformly throughout.

**Response:** Thank you for this suggestion. We will keep checking such inconsistencies when revising the manuscript later.

**References mentioned in the responses:**

Jaiswal, K. and Wald, D. J.: Rapid estimation of the economic consequences of global earthquakes, US Department of the Interior, US Geological Survey Reston, VA, 2011.

Li, B.: Comparative Analysis of Estimates on Capital Stock of China, Journal of Quantitative and Technological Economics, 21–54, https://doi.org/10.13653/j.cnki.jqte.2011.12.006, 2011.

Wu, J., Li, N., and SHI, P.: Benchmark wealth capital stock estimations across China's 344 prefectures: 1978 to 2012, China Economic Review, 31, 288–302, https://doi.org/10.1016/j.chieco.2014.10.008, 2014.

Xin, D., Daniell, J. E., Tsang, H.-H., and Wenzel, F.: Residential building stock modelling for mainland China targeted for seismic risk assessment, Nat. Hazards Earth Syst. Sci., 21, 3031–3056, https://doi.org/10.5194/nhess-21-3031-2021, 2021.

---

## Author Response (AR1)

The manuscript outlines a methodology for constructing a time-dependent fixed asset model applicable to China. It establishes a model using fixed asset data at the provincial level and further refines it to a more detailed grid level with remote sensing ancillary data. The findings are validated with results from other studies. The methodology is purported to assist in quick post-earthquake loss estimations. The topic is interesting, and the manuscript is well-written. However, it can benefit from some comments I suggest below to improve its clarity and contribution to the field.

**Response:** Thank you very much for your time and efforts spent on reviewing our manuscript, which is deeply appreciated. We have read all your constructive comments and suggestions carefully and tried our best to prepare the following responses. Hopefully, they can help release the concerns you have on the prior version of the manuscript.

**General comment:**

The methodology and results seem useful for assessing economic losses due to any hazard, not only due to earthquakes. Even though the introduction includes references about the importance of earthquakes and previous earthquake loss estimations, the methodology and results do not confront loss estimations from past earthquake events. Furthermore, Fig. 2 has no input that one could identify with an earthquake hazard or risk (e.g., ground motion fields, earthquake occurrence rate, or fragility model). The methodology and results focus on modelling the observed economic and demographic growth in the study areas consistently, considering the data quality limitations in terms of spatial resolution and time-frequency. I think the paper can be stronger (and closer to the journal's scope) by adding some statements in the discussion and conclusion about how the methodology and results can help in the loss estimation due to an earthquake event or other natural hazards. It would also be interesting to add to the introduction a review of studies of other natural hazards, such as floods or extreme wind events.

**Response:** Thank you for this comment and all your suggestions. To better demonstrate the application of the developed fixed asset data to seismic loss estimation, we have added Section 4.3 and Figure 13 to the revised manuscript indicating how the fixed asset data can be overlapped with the macro-seismic intensity map for the Ms6.2 Jishishan earthquake occurred on December 18, 2023 in Gansu province, China, as given in **Lines 514-540, Pages 23-25** of **newly added Section 4.3** in the clean version of the revised manuscript.

We agree that it is also necessary to have an in-depth discussion on the applicability of the developed fixed asset data to the risk analysis of other natural hazards (e.g., flood, wind, etc.). We have added this discussion to **Lines 541-559, Page 25** in the **newly added**

**Section 4.3** of the revised manuscript.

For the suggestion to add a review of other natural hazards in the Introduction section, considering the focus of this paper is to introduce how the fixed asset model is developed and disaggregated, the application of this fixed asset data to seismic loss estimation is added and its limitation for risk analysis of other natural hazards is also discussed in-depth in the revised manuscript, we think adding an extensive review of other natural hazards (e.g., flood, wind etc.) is not that closely related to the focus of this paper.

In addition, there are already quite good review articles written by specialists focusing on risk analysis of flood and wind. Among them, De Moel et al. (2015) provided a quite comprehensive overview on the current state, development, assessment characteristics of quantitative flood risk assessments at different scales (supra-national, macro, meso, micro). They also outlined the lessons learnt from current practice and identified future research needs for flood risk assessment. Yu et al. (2023) gave a comprehensive review on the studies related to exposure roughness exposed to wind hazard as well. Therefore, we consider it is better not to give an extensive but not in-depth overview of other natural hazards in the Introduction section that is centered around the earthquake hazard.

**Specific comment:**
1. The conclusion says the methodology can be extended to more recent years once the data is available. However, considering that the methodology aims to help in quick loss assessment for future earthquake events, can the methodology with the available data today (in 2024) provide a prediction, for example, of the losses after an earthquake event in 2030?
   **Response:** Thank you for this question. The direct answer is yes if prediction accuracy is not strictly required. And the accuracy can be enhanced if the increased fixed asset values from 2024 to 2030 are available as well. Rapid seismic loss estimation after the occurrence of a damaging earthquake is based on the combination of intensity map, fixed asset in the earthquake-stricken area, and the empirically regressed vulnerability model for the earthquake-affected region. For an earthquake to occur in 2030, we would recommend the use of latest fixed asset map available to assess its seismic loss, as long as the intensity map can be reasonably modelled/predicted by using ground motion prediction equations or physics-based simulation methods, and the vulnerability curve can be rectified by considering the changing in building vulnerability from 2020 to 2030 as well, since currently the available vulnerability curve developed in our another work (Li et al., 2023) is only based on damaging information of

earthquakes occurred before 2020.

2. I suggest mentioning the future availability of grid-level fixed asset data only in the section "code and data availability", as it is done and justified in this version of the manuscript, and only mentioning it in the abstract and conclusion in a later version, when the data is effectively available.

   **Response:** Thank you for this suggestion. The expression related to data availability in the abstract and conclusion has been removed in the revised manuscript.

**Technical corrections**

1. Tables 2 and 3, as well as Figs. 5, 7-9: Consider changing the monetary units to billion Chinese yuans, as done in Figs. 11-12

   **Response:** Thank you for this suggestion. In Tables 2 and 3 of the revised manuscript, the monetary units of numbers (representing the sum of fixed asset value in multiple cities and provinces) have been changed to billion Chinese yuan. However, in Figs. 5.,7-9, the value represents the fixed asset in each 1km×1km grid, not like that in city agglomerations or provinces in Tables 2 and 3. Therefore, to better demonstrate the spatial location of high fixed asset clusters, the upper threshold of fixed asset value in the color bar of Fig. 5, Fig. 7, Fig. 8, and Fig. 9 is automatically set as 243801864, 74247334, 61145222, and 88179937 in QGIS, respectively, which corresponds to the 98% quantile of the grid-level fixed asset value in each figure. These upper threshold values are all smaller than 1 billion since these values only represent fixed asset in those 1km×1km grids. If the unit in these figures is expressed in billion Chinese yuan, the numbers at grid level will be too small to differentiate the spatial clusters of high fixed assets (since the upper threshold values in Figs. 5-8 and Figure 9 will be 0.244, 0.074, 0.061, and 0.088 billion Chinese yuan respectively). Therefore, for better visualization effect and the comparison among different urban agglomerations, using the monetary unit of yuan for the automatically determined thresholds corresponding to 98% quantile of grid-level fixed asset in each figure is a relatively better choice.

2. Fig. 2: There is a typo in one of the charts: "Harmonized" instead of "Harmanized".

   **Response:** Thank you so much for your careful check! This error has been rectified in Figure 2 of the revised manuscript (in **Line 155, Page 6**).

3. Although described in the text, the delta in Eq. 3 has a different meaning than the delta in Eq. 4. I suggest using a different symbol for one of them.

   **Response:** Thank you so much for this suggestion! We has used $\kappa$ to replace $\delta$ in Eq. (3) and in related context of the revised manuscript (in **Lines 229-230, Page 9**).

4. Table 1: There is a typo in the 2nd column, 8th row: "Population density data" instead of "Population dentsity data".

Response: Thank you again for your so careful check! This typo has been rectified in the revised manuscript (in **Line 156, Page 6**).

**RC2 comments and responses**

**The overall quality of the preprint (general comments)**

The overall quality of the paper is high. The topic of developing a novel fixed asset model to improve seismic loss estimation is significant for the science community. By mapping fixed assets at a 1 km × 1 km grid level, the model better serves rapid seismic loss assessments and informs emergency response plans. The research is well structured and explained. The authors made an effort to combine various data sources and techniques. While the model represents a significant advancement, several limitations could impact its effectiveness, particularly in high-stakes applications like earthquake response. The paper's scientific contributions justify publication with minor revisions to handle specific data assumptions better and further validate the model's application. These adjustments would help ensure the model's broader applicability and robustness.

**Response:** We deeply appreciate the time and efforts you have devoted to improving the quality of this manuscript and thank you so much for all the constructive comments! Our detailed responses are given as follows. Hopefully they can help release your concern on the earlier version of the manuscript.

**Individual scientific questions/issues (specific comments)**

Here are some topics which the authors could discuss in more detail:

**Reliance on Historical Investment Data and Simplified Depreciation Rates**. The model bases its estimates on historical investment data and applies a uniform 5% depreciation rate across all provinces, regardless of variations in asset type, economic condition, or regional maintenance practices. The uniform deprecation rate can introduce inaccuracies, especially for assets with different service lives or conditions. The simplified approach to depreciation may lead to skewed asset values, particularly in provinces with unique economic trajectories or asset compositions. For instance, in industrialized regions, assets may have a shorter useful life than in less industrialized provinces, affecting the accuracy of economic loss projections. Can the model be refined by including a variable depreciation rate based on more detailed asset-specific and regional data, if available?

**Response:** Thank you very much for this pertinent suggestion! We totally agree that it is quite necessary to integrate the temporal and spatial change in depreciation rate when modelling the net value of depreciated fixed asset, should the statistical data required to differentiate such rates be accessible for the period 1951-2020 considered in this study. As a matter of fact, in a prior prefecture-level fixed asset modeling work of Wu et al. (2014) for China during 1978-2012, they did develop varying depreciation rates for different provinces.

We have added a **new section 4.4** "Limitations of the modelled fixed asset data" in the clean version of the revised manuscript (see **Lines 560-604, Pages 25-27**). In this section, a more detailed introduction on how the provincial level depreciation rates in Wu et al. (2014) are derived is given. In addition, a new figure (**Figure 14** in **Lines 591-593** of the revised manuscript) comparing the fixed asset change ratio at provincial level by using the depreciation rates in Wu et al. (2014) and by using a fixed rate of 5% and related discussion is also given.

**Inconsistencies in the Data Sources for Ancillary Datasets.** The model relies on ancillary datasets (e.g., nighttime lights, population, built-up areas) which are not consistently available across all years. This results in the use of alternative data types to approximate missing data. For instance, population data alone is used in the early years when nighttime light data is unavailable. These proxies may not accurately represent economic activity, especially in rural areas or less-developed regions, leading to potential over- or under-estimations in asset distribution. Could the model be strengthened by incorporating more recent, high-resolution satellite data or by exploring alternative disaggregation methods that do not depend solely on proxies like nighttime lights?

**Response:** Thank you for this comment. A big effort made in this paper is to find reasonable combination of different ancillary data to disaggregate provincial level fixed asset into grid level. For periods (1991-2020) when nighttime light data are available, the combination of nighttime light and population are used to create the lit-pop index, which is exactly to better avoid the over- or under-estimation problems in asset distribution by using nighttime light or population data alone. For years before 1991 (1971-1990), when nighttime data are unavailable, the built-up surface area data and population data are used to create the area-pop index. And for earlier periods (1951-1970) when only grid level population density data are available, we choose to apply the pop-pop index (derived from the squared value of population in each 1km×1km grid) to further disaggregate the asset value of years before 1970. And the correlation analysis in Figure 10 between each pair of three disaggregation indexes (lit-pop, area-pop, pop-pop) further validates the consistency among these indexes.

But for earlier periods (1951-1970) when only population density data are used to disaggregate the provincial level fixed asset, we consider it is not appropriate to incorporate the high-resolution satellite data in recent years with population data to disaggregate the asse data. The reason is that China has experienced quite different economic development stages before 1970 and nowadays. While before 1970, the economy development in China was very slow due to natural disaster, political movement and the planned economic system, thus the fixed asset distribution pattern cannot be projected by using recent

remote-sensing data.

**Lack of Structural Detail in Asset Composition.** The model's focus on fixed capital should differentiate between asset types (e.g., residential vs. industrial buildings) in disaggregation. This lack of structural specificity reduces the model's utility for applications that require asset type differentiation, such as insurance underwriting or infrastructure resilience planning. Different asset types respond differently to seismic events; for instance, infrastructure like bridges and roads may sustain different levels of damage compared to residential buildings. This generalization could lead to misaligned resource allocations during emergency responses. Introducing asset type categorization, possibly by incorporating land use or building inventory data, would enhance the model's accuracy for specific asset loss estimations.

**Response:** We totally agree with this comment. If the research focus is to model the fixed asset value for specific years only, it is possible to use more detailed census or even in-site investigated data to estimate the value of different asset types and disaggregate each asset type into grid level by using different remote sensing ancillary data as proxy (e.g., Gunasekera et al., 2015; Wu et al., 2018; Xin et al., 2021).

However, when accumulated fixed asset data series for a long period dating back to 1951 are needed, as modelled in this paper, detailed statistics to differentiate the building types (residential/industrial/commercial) and even the quota of different fixed assets (buildings, infrastructures, instruments) exposed to past years are typically missing. In this case, the fixed asset model developed in this paper has to be based on the Level 1 data. Based on this level information, the estimated seismic loss is relatively a rough estimation since the input data mainly include demographic data and capital stock investment data extracted from the yearbooks or national census. And when seismic loss is rapidly estimated after the occurrence of a damaging earthquake, the vulnerability curve to be used is also a quite empirical one, similar to those developed in Jaiswal and Wald (2011) in the PAGER (Prompt Assessment of Global Earthquakes for Response) project. Such empirical vulnerability curve only describes the relation between the mean loss ratio and the macro intensity, which is regressed from damaging statistics (intensity map, total loss, exposed asset value) of historical earthquakes for specific countries and regions. And such loss estimation process is quite different from the one based on the structure type and apply the corresponding vulnerability curves for specific buildings.

Although in the fixed asset data developed in this paper, there is no differentiation of building types, when compared the modelled asset with our previous work focusing on developing the replacement value of residential buildings and structures (Xin et al., 2021),

as shown in Figure 12, we find the correlation ratio at prefecture level between these two asset models are relatively high (with $R^2$=0.91). It is noteworthy that asset models with detailed building attributes for specific year only is not sufficient to develop the empirical vulnerability curve, which entails the fixed asset for past years as well. Considering the completeness and availability of input data for period 1951-2020, the perpetual inventory method (PIM) is the best practice to follow to develop such asset data series.

The above discussion and part of your comments have been further briefly summarized and added to the **new section 4.4** "Limitations of the modelled fixed asset data" in **Lines 606-621, Pages 27-28** of the revised manuscript.

**Technical corrections**

The article's language quality is overall sound, with a few areas where readability and formality can be improved. Here are specific suggestions regarding grammar, spelling, and phrasing.

- Maintain past tense in descriptions of the completed study. For example, in the sentence, "The fixed asset model to be developed in this paper is also based on the Level 1 data."

  **Response:** Thank you for pointing this out. We have thoroughly checked the use of verb tenses when revising the manuscript. However, we have to confess that sometimes it is kind of confusing to decide which tense should be used. For example, the expression "*The fixed asset model to be developed in this paper is also based on the Level 1 data*" can also be regarded as a description of the fact. In this case, it seems the present tense should be used, as suggested by the book "*Science Research Writing ---- For Non-Native Speakers of English*" (Glasman-Deal, 2010).

- Remove redundant phrases. For example, "To summarize, the nighttime light data and GHS-POP data are used to generate the lit-pop disaggregation indexes from 1991 to 2020, while the GHS-BUILT-S data and GHS-POP data are used to construct area-pop disaggregation indexes from 1971 to 1990..."

  **Response:** Thank you very much for your careful check. The initial consideration to give such summarization in the Results section is mainly to help readers who only roughly scan this paper to quickly understand the difference in ancillary data used for different periods when disaggregating the provincial fixed asset value. Therefore, we consider it might be better to keep such a reminder.

- Avoid Informal Language. For example, replacing "Luckily" with "Fortunately" is more formal.

**Response:** Thank you very much for this suggestion! We have reread the manuscript carefully and replaced the expression like "Luckily" to "Fortunately" accordingly. In addition, we also used the software Grammarly to check the whole revised manuscript, to help make sure that all the expressions are correctly organized. For example, "greatly" has been replaced by "significantly", "next" has been replaced by "following", and "huge" has been replaced by "considerable", etc.

- Revise for Consistency In Abbreviations and Acronyms. Introduce abbreviations consistently upon first mention, ensuring they are used uniformly throughout.
  **Response:** Thank you for this reminding! With the help of Grammarly and self-crosscheck, we have tried our best to avoid such inconsistencies when revising the manuscript. For example, the expressions like "fixed capital stock, fixed asset" have been uniformly expressed as "fixed asset" in the revised manuscript.

**References mentioned in the responses:**

De Moel, H., Jongman, B., Kreibich, H., Merz, B., Penning-Rowsell, E., and Ward, P. J.: Flood risk assessments at different spatial scales, Mitig Adapt Strateg Glob Change, 20, 865–890, https://doi.org/10.1007/s11027-015-9654-z, 2015.

Glasman-Deal, H.: Science research writing for non-native speakers of English, World Scientific, 2010.

Gunasekera, R., Ishizawa, O., Aubrecht, C., Blankespoor, B., Murray, S., Pomonis, A., and Daniell, J.: Developing an adaptive global exposure model to support the generation of country disaster risk profiles, Earth-Science Reviews, 150, 594–608, 2015.

Jaiswal, K. and Wald, D. J.: Rapid estimation of the economic consequences of global earthquakes, US Department of the Interior, US Geological Survey Reston, VA, 2011.

Li, Y., Xin, D., and Zhang, Z.: Estimating the economic loss caused by earthquake in Mainland China, International Journal of Disaster Risk Reduction, 95, 103708, https://doi.org/10.1016/j.ijdrr.2023.103708, 2023.

Wu, J., Li, N., and SHI, P.: Benchmark wealth capital stock estimations across China's 344 prefectures: 1978 to 2012, China Economic Review, 31, 288–302, https://doi.org/10.1016/j.chieco.2014.10.008, 2014.

Wu, J., Li, Y., Li, N., and Shi, P.: Development of an asset value map for disaster risk assessment in China by spatial disaggregation using ancillary remote sensing data, Risk analysis, 38, 17–30, 2018.

Xin, D., Daniell, J. E., Tsang, H.-H., and Wenzel, F.: Residential building stock modelling for mainland China targeted for seismic risk assessment, Nat. Hazards Earth Syst. Sci., 21, 3031–3056, https://doi.org/10.5194/nhess-21-3031-2021, 2021.

Yu, J., Stathopoulos, T., and Li, M.: Exposure factors and their specifications in current wind codes and standards, Journal of Building Engineering, 76, 107207, https://doi.org/10.1016/j.jobe.2023.107207, 2023.

---

## Referee Report (RR1)

**COMMENTS** for manuscript nhess-2024-138

**1. The overall quality of the preprint (general comments)**

The revised manuscript presents significant improvements over the initial submission. The authors have effectively addressed several key concerns raised during the review process, particularly in refining their methodology, providing additional validation, and clarifying the model's applications. The inclusion of Sections 4.3 and 4.4, along with new figures (Figures 13, and 14), has notably enhanced the manuscript's clarity and robustness. The study remains a valuable contribution to seismic loss estimation and exposure modeling, with its detailed grid-level fixed asset model providing a novel and practical dataset for China's disaster risk assessment community.

However, despite these improvements, some minor revisions are still recommended to further strengthen the manuscript before final acceptance. The remaining concerns primarily pertain to depreciation rate variability, early-year data reliability, and asset categorization, all of which could enhance the study's applicability and methodological robustness with additional refinements

**2. Individual scientific questions/issues (specific comments)**

Here are some areas for minor revision:

**Depreciation Rate Variability**

While the authors acknowledge the importance of variable depreciation rates, the model continues to use a fixed 5% rate across all provinces and time periods. This assumption may not fully capture regional differences in asset longevity, economic conditions, and maintenance practices.

Suggested Revision: Include a sensitivity analysis testing different depreciation rates (e.g., 5%, 7%, and 9%) and their impact on estimated asset values. This would provide a clearer understanding of the model's robustness and potential variations in loss estimation.

Even if the dataset constraints prevent full integration of variable rates, a quantitative discussion on the potential impact of using different rates would add credibility to the model.

**Early-Year Data Reliability**

The authors justify their approach of using population density data as a proxy in the absence of nighttime light or built-up area data before 1971. While reasonable, this introduces uncertainty in the fixed asset estimates for early years.

Suggested Revision: Provide a comparative analysis of early-year fixed asset estimations against alternative historical economic indicators (e.g., historical GDP, industrial output). If

direct comparisons are not possible, a brief discussion of potential error margins in early-year estimations would help.

Addressing possible inconsistencies in early-year estimates will strengthen confidence in the model's historical reliability, particularly for long-term economic trend analysis

**Asset Type Categorization**

The model does not differentiate between asset types (residential, commercial, industrial), limiting its utility for applications that require specific asset vulnerability assessments.

Suggested Revision: If differentiation is not feasible due to data limitations, clarify in the discussion how future studies could incorporate land-use data, building inventories, or economic sector data to refine the model.

Even if asset-type differentiation is beyond the scope of this study, acknowledging its importance and outlining potential future directions would improve the paper's practical relevance.

**3. Technical corrections**

The authors have made commendable efforts in refining the manuscript's clarity, addressing most grammatical and typographical errors. However, the following minor adjustments are recommended (disregard if this reviewer saw it wrongly):

- Equation Consistency: The symbol for delta ($\delta$) is used with different meanings in Equations (3) and (4). Consider replacing one of them to avoid confusion.
- Units in Figures 5, 7-9: While the justification for using monetary units at the grid level is reasonable, adding a brief explanatory note in the figure captions would help readers understand why values differ from Tables 2-3.

**4. Conclusion and Final Recommendation**

The manuscript is in near-final form and has substantially improved through this revision process. The remaining concerns are minor and can be addressed through a final revision focusing on sensitivity analysis for depreciation rates, historical data validation, and an expanded discussion on asset categorization. These changes will further solidify the paper's contributions to seismic risk modeling and disaster economics.

---

## Author Response (AR2)

Dear Danhua and co-authors,

For publication, I kindly ask you to review the referee reports and address their comments and revisions. Their suggestions are shown below and in the attached report. Please provide me with a revised version and mark the changes you make.

Thank you.

Solmaz Mohadjer

NHESS Editor

**Response:** Dear Editor Solmaz, thank you very much for helping to manage the whole review process! We have revised the manuscript according to the suggestions from both referees. The point-to-point responses are given below.

**Report 1:**

L515: The sentence is not clearly written (Since the development of ..., which is of vital reference importance..., then what?). Please correct it.

**Response:** Accepted! Behind this sentence "*Since the development of the fixed asset data is targeted for seismic loss estimation after the occurrence of a damaging earthquake in China, which is of vital reference importance for government officials to reasonably allocate emergency response personnel and goods as well as for insurance and reinsurance companies to quickly estimate the potential compensation amount*", the expression "*it is necessary to check the application potential of the developed dataset.*" has been added to **Line 518, Page 24** of the revised manuscript (clean version). Thank you for your careful check!

**Report 2:**

**Comment 1:** Here are some areas for minor revision: Depreciation Rate Variability - While the authors acknowledge the importance of variable depreciation rates, the model continues to use a fixed 5% rate across all provinces and time periods. This assumption may not fully capture regional differences in asset longevity, economic conditions, and maintenance practices. *Suggested Revision: Include a sensitivity analysis testing different depreciation rates (e.g., 5%, 7%, and 9%) and their impact on estimated asset values.* This would provide a clearer understanding of the model's robustness and potential variations in loss estimation. Even if the dataset constraints prevent full integration of variable rates, a quantitative discussion on the potential impact of using different rates would add credibility to the model.

**Response:** Thank you for further emphasizing this issue. When the depreciation rate is fixed as 7% and 9%, ratios between provincial-level fixed assets relative to that with the depreciation rate fixed as 5% is shown in the following Figure R1 and Figure R2, respectively. The changing trends of fixed assets with time shown in Figures R1-R2 are quite similar to that in Figure 14 of the manuscript

(which has been slightly revised in this version due to typos and description change in Y axis). For cross-check convenience, we also paste Figure 14 as Figure R3 here. In Figure 14, sensitivity test has been conducted by using different depreciate rates (ranging from 7.95% to 10.05%, as listed in Table 4 of the manuscript) developed by Wu et al. (2014) for each province. Discussions on this sensitivity test and the changing trend in Figure 14 have been given in **Lines 561-600, Page 27** of the manuscript (clean version). Due to the similarity between Figures R1-R2 and Figure 14, we prefer not to further repeat such discussion.

[Figure]

**Figure R1:** Ratios between provincial-level fixed assets with depreciation rates set as **7% and 5%**, respectively.

[Figure]

**Figure R2:** Ratios between provincial-level fixed assets with depreciation rates set as **9% and 5%**, respectively.

[Figure]

**Figure R3:** Figure 14 of the manuscript, which demonstrates the ratio between provincial-level fixed assets with varying depreciation rates developed by Wu et al. (2014) and the fixed rate of 5%.

**Comment 2:** Early-Year Data Reliability - The authors justify their approach of using population density data as a proxy in the absence of nighttime light or built-up area data before 1971. While reasonable, this introduces uncertainty in the fixed asset estimates for early years. *Suggested Revision: Provide a comparative analysis of early-year fixed asset estimations against alternative historical economic indicators (e.g., historical GDP, industrial output). If direct comparisons are not possible, a brief discussion of potential error margins in early-year estimations would help.* Addressing possible inconsistencies in early-year estimates will strengthen confidence in the model's historical reliability, particularly for long-term economic trend analysis.

**Response:** This is a quite good suggestion. Currently, a comparison between modelled provincial-level fixed assets and GDP has been conducted and shown in Figure 4 of the manuscript. As indicated by Figure 4, the ratios between provincial-level fixed assets and GDP appears to be much irregular in early periods due to the lack of an official and standard recording method at that time. The discussion on such irregularity has been given in **Lines 356-370, Pages 13-14** of the manuscript. In **Lines 181-185, Page 7**, we also introduced the sensitivity tests by changing early-year asset estimation conducted by previous studies, namely "*Fortunately, previous studies (Shan, 2008; Wu et al., 2014; Zhang et al., 2004) have demonstrated that the effect of the initially determined fixed asset value of the base year on the asset estimation for the following years will decline given sufficiently long time series. For example, the sensitivity test performed by Wu et al. (2014) indicated that a doubling of the initial asset value in 1978 only resulted in less than a 0.6% change in the stock estimation in 2012.*".

In addition, the following discussion related to grid-level fixed asset evaluation and application will also be added to **Lines 605-612, Pages 27-28** of the revised manuscript:

"*For evaluation of the grid-level fixed asset model disaggregated from provincial-level data in the early periods, a direct comparative analysis with related statistical records would be quite valuable. However, such comparison is unfortunately hindered by the lack of prefecture/county level statistical records of GDP or industrial output data in the early periods, as can be checked from the official website of the National Bureau of Statistics (https://data.stats.gov.cn/english/). Therefore, considering the rough estimation made in the determination process of the initially accumulated fixed assets as well as the lack of an official and standard method in the compilation of economic indicators in the early periods after 1949, special attention should be paid when applicating the developed grid-level fixed asset data to studies like long-term economic trend analyses for specific regions.*"

**Comment 3:** Asset Type Categorization - The model does not differentiate between asset types (residential, commercial, industrial), limiting its utility for applications that require specific asset vulnerability assessments. *Suggested Revision: If differentiation is not feasible due to data limitations, clarify in the discussion how future studies could incorporate land-use data, building inventories, or economic sector data to refine the model.* Even if asset-type differentiation is beyond the scope of this study, acknowledging its importance and outlining potential future directions would improve the paper's practical relevance.

**Response:** Accepted! In the revised manuscript, the following discussion will be added to **Lines 626-631, Page 28** to further specify the model improvement directions:

"*In the future, to better aid the natural disaster risk assessment and emergency response needs, more efforts should be made to employ more remote sensing data as well as detailed building and infrastructure related statistics to refine the latest fixed asset model. For example, land-use data can be integrated into the modelling process to better discriminate land types and exposed elements, detailed building-related census records can be adopted to better quantify the asset share of different building structures, and road density data can be used to well improve the asset evaluation and disaggregation accuracy of infrastructures.*"

---

## Author Response (AR3)

Dear Danhua Xin, James Daniell, Zhenguo Zhang, Friedemann Wenzel, Shaun Wang, and Xiaofei Chen,

I am pleased to inform you that the revised version of your manuscript NHESS-2024-138 can now be accepted for publication in our journal Natural Hazards and Earth System Sciences (NHESS). Please inform your co-authors of the editorial decision. However, I would like to request some technical corrections before final publication, please see below:

**Response:** Dear Editor Solmaz, we want to express our sincere gratitude to you for your careful and patient review-managing efforts! We also appreciate the correction points you mentioned and have prepared our point-to-point responses, as given below.

-Fig 2 - Please define all abbreviations used in this figure in the caption and indicate the reading direction by using arrows if needed. In the caption, please explain the yellow vs white sections. Was this done to differentiate steps in the workflow?

**Response:** Thank you for this reminding. Yes, the boxes marked in different colors are to differentiate the two main parts in the workflow. The following caption has been added to **Lines 156-164, Page 6** in the revised manuscript:

"*Boxes are marked by different colors to differentiate the two main parts in the workflow. The boxes marked in yellow refer to the steps required to develop the provincial-level fixed asset data, and the boxes marked in white are components used to construction the disaggregation indices (see Section 2 for more details). The indices "lit-pop, area-pop, pop-pop" refer to the disaggregation index generated using nighttime light and population, build-up area and population, and population data alone, respectively. The abbreviation "DMSP/OLS" refers to nighttime light observations acquired by the US Air Force Defense Meteorological Satellite Program (DMSP) Operational Linescan System (OLS), "VIIRS" refers to nighttime light observations from the Visible Infrared Imaging Radiometer Suite instrument, "GHS-BUILT-S" refers to the built-up surface data provided by the Global Human Settlement Layer project, and "GHS-POP" refers to the population density data provided by the same project.*"

-Fig 3 - Please explain the marking of the years in figure caption. Do you mean the bottom line marks the year 1951? If so, please state that in the caption.

**Response:** Yes, exactly. The statement "*The bottom line marks the accumulated fixed asset in year 1951 and the top line marks the accumulated fixed asset in year 2020.*"

has been added to the caption of Figure 3 in **Lines 380-381, Page 14** of the revised manuscript.

-Fig 13 - Please include an inset map of China (showing the whole country and the area of interest in Fig 13).
**Response:** Accepted! Figure 13 has been revised to include the inset map of China in **Line 546, Page 26** of the revised manuscript.

-Table 4 - If this table has been modified from the Table 4 of Wu et al. (2014), please indicate this in the caption. Otherwise, it might be enough just to keep the reference in the text and remove the table from your manuscript.
**Response:** Thank you for this verification. Table 4 in this paper is modified from Table 2 in Wu et al. (2014). We have specified this in **Line 590, Page 27** of the revised manuscript.

-Line 180 - Revise to 'The determination process of the initially accumulated fixed asset value for the base year has inevitable uncertainty.'
**Response:** Accepted!

-Line 190 - Delete 'It is worth notifying that'
**Response:** Accepted!

-Line 227 - Revise to 'including but not limited to construction and installation, equipment and instrument'
**Response:** Accepted!

-Line 546 - By 'wind', do you mean 'storm surges' or 'high/extreme winds'?
**Response:** Thank you for this reminding. We have specified "wind" as "high/extreme wind" in **Lines 551 and 564, Page 26** of the revised manuscript.

-Line 554-555 - Informal language, consider revision.
**Response:** This sentence has been modified as "*In contrast, Bouwer et al. (2009) found that using a 100-m instead of a 25-m exposure grid in loss estimation would result in increased flood damage estimates up to 50% higher.*" in **Lines 563-564, Page 26** of the revised manuscript.

-Is there a reason for referring to the Chinese currency using two different terms (CNY

vs RMB)? If yes, please explain. If not, I suggest sticking with one term to avoid confusion.

**Response:** Thank you for your careful check. There is no difference between them and "RMB" has been changed to "CNY" in **Line 31, Page 2** of the revised manuscript.

-Please replace all 'natural disaster' terms with 'disasters'. It misleads readers to think the devastating results are simply part of a natural process which is not true.

**Response:** Thank you for reminding the potential misleading. Compared with deleting "natural" in this expression, we consider replacing "natural disasters" with "natural hazards" can better reflect the hazard type we focus in this paper and also avoid the possible misleading.

You will soon be contacted by the journal publication office for the preparation of the final version of your manuscript. Please respond promptly to the requests of the publication office, as this will condition the time of your publication, and read carefully any copy-editing recommendations that they make, in addition to the changes requested above. Let me remind you that final publication of your paper on the journal web site is subject to the payment of the appropriate service charges. Also, do not forget that as an open-access journal you can freely distribute your paper once it is published.

**Response:** Thank you for this reminding! We will get the payment done as soon as we receive the bill and respond to all the recommendations from the publication office properly and promptly. And we do hope that the publication could be made as early as possible.

I take this opportunity to thank you and your co-authors for having selected Natural Hazards and Earth System Sciences (NHESS) for the publication of your scientific work. I look forward to receiving further contributions to our journal from you and your co-authors.

**Response:** We want to thank you again, dear Editor Solmaz, for all the efforts you have made during the review process, which is quite impressive and touching. We also sincerely wish the NHESS journal will be better and better with the efforts of you and all the colleagues. Best wishes!

Best,
Solmaz Mohadjer
NHESS Editor